# An Extractive Electrospray Ionization Time-of-Flight Mass Spectrometer (EESI-TOF) for online measurement of atmospheric aerosol particles.

Felipe D. Lopez-Hilfiker[1,a], Veronika Pospisilova[1], Wei Huang[2], Markus Kalberer[3,4], Claudia Mohr[5], Giulia Stefenelli[1], Joel A. Thornton[6], Urs Baltensperger[1], Andre S. H. Prevot[1], Jay G. Slowik[1]

[1]Laboratory of Atmospheric Chemistry, Paul Scherrer Institute (PSI), 5232 Villigen PSI, Switzerland

[2]Institute of Meteorology and Climate Research, Karlsruhe Institute of Technology, 76131 Karlsruhe, Germany

[3]Department of Chemistry, University of Cambridge, Cambridge CB2 1EW, United Kingdom

[4]Department of Environmental Sciences, University of Basel, 4056 Basel, Switzerland

[5]Department of Environmental Science and Analytical Chemistry, Stockholm University, 106 91 Stockholm, Sweden

[6]Department of Atmospheric Sciences, University of Washington, Seattle, WA 98195, Washington, USA

[a]*Now at*: Tofwerk AG, 3600 Thun, Switzerland

*Correspondence to*: Jay G. Slowik (jay.slowik@psi.ch)

**Abstract**

Real-time, online measurements of atmospheric organic aerosol (OA) composition are an essential tool for determining the emissions sources and physicochemical processes governing aerosol effects on climate and health. However, the reliance of current techniques on thermal desorption, hard ionization, and/or separated collection/analysis stages introduces significant uncertainties into OA composition measurements, hindering progress towards these goals. To address this gap, we present a novel, field-deployable extractive electrospray ionization time-of-flight mass spectrometer (EESI-TOF), which provides online, near-molecular (i.e., molecular formula) OA measurements at atmospherically relevant concentrations without analyte fragmentation or decomposition. Aerosol particles are continuously sampled into the EESI-TOF, where they intersect a spray of charged droplets generated by a conventional electrospray probe. Soluble components are extracted, and then ionized as the droplets are evaporated. The EESI-TOF achieves a linear response to mass, with detection limits on the order of 1 to 10 ng m$^{-3}$ in 5 s for typical atmospherically-relevant compounds. In contrast to conventional electrospray systems, the EESI-TOF response is not significantly affected by a changing OA matrix for the systems investigated. A slight decrease in sensitivity in response to increasing absolute humidity is observed for some ions. Although the relative sensitivities to a variety of commercially available organic standards vary by more than a factor of 30, the bulk sensitivity to SOA generated from individual precursor gases varies by only a factor of 15. Further, the ratio of compound-by-compound sensitivities between the EESI-TOF and an iodide adduct FIGAERO-CIMS vary by only ±50%, suggesting that EESI-TOF mass spectra indeed reflect the actual distribution of detectable compounds in the particle phase. Successful deployments of the EESI-TOF for laboratory environmental chamber measurements, ground-based ambient sampling, and proof-of-concept measurements aboard a research aircraft highlight the versatility and potential of the EESI-TOF system.

## 1. Introduction

Aerosol particles adversely affect respiratory and cardiovascular systems, scatter and absorb radiation, influence cloud formation processes and properties, provide surfaces for heterogeneous reactions, and affect trace gas concentrations by providing an adsorptive medium for semi-volatile gases. As a result, aerosols have a significant effect on public health, climate, and overall atmospheric reactivity. Of particular importance are aerosol particles smaller than 1 µm in diameter, a significant and ubiquitous fraction of which is secondary organic aerosol (SOA) formed from atmospheric reactions of organic gases (Hallquist et al., 2009; Jimenez et al., 2009). The sources, aging, and chemical properties of SOA remain highly uncertain, and these uncertainties can lead to large errors between modeled and measured aerosol loadings (Volkamer et al., 2006). These errors limit our ability to predict future changes in aerosol particle composition and concentration under a warming climate and to link SOA to its atmospheric emission sources. To develop adequate model parameterizations of organic aerosol (OA) and its formation, growth, and loss, there remains a need to improve source apportionment capabilities and to identify chemical mechanisms governing the conversion and partitioning of organic compounds between gas and particle phase. Measurements of specific chemical tracers on timescales similar to the typical variability in emissions, photochemical activity, and meteorology, approximately minutes to hours, would improve source apportionment, mechanistic studies, and characterization of bulk molecular properties such as the distribution of average oxidation state across carbon number (Kroll et al., 2011) against mechanistic photochemical models.

The chemical complexity of OA makes highly time-resolved, chemically-specific measurements extremely challenging. OA consists of thousands of individual components, many of which are present at only trace amounts (Goldstein and Galbally, 2007). Most organic aerosol chemical speciation to date has come from offline analysis of filter samples. Offline samplers typically concentrate particles on a filter or impactor for 2-24 hours, after which particles are extracted or thermally desorbed for offline chemical analysis (Hallquist et al., 2009). The main disadvantages of offline techniques are their low time resolution, which is much slower than many atmospheric processes, and the potential for chemical changes due to evaporation, adsorption, and/or reaction during sample collection, transfer, and/or storage (Turpin et al., 1994; Subramanian et al., 2004; Timkovsky et al., 2015; Kristensen et al., 2016). The extraction process is also time consuming and can introduce additional artifacts into the measurements (e.g. decomposition during

derivatization or hydrolysis in aqueous solutions). A recent study addresses the issue of low time resolution and reaction on the collection substrate by collecting samples at 5 min time resolution using a particle-into-liquid sampler (PILS) coupled to collection vials on a rotating carousel, followed by offline ultra-performance liquid chromatography/electrospray ionization quadrupole

time-of-flight mass spectrometry (UPLC/ESI-Q-TOFMS) analysis (Zhang et al., 2015; Zhang et al., 2016). This approach yielded time-resolved molecular speciation for water-soluble components, but remains subject to sample transfer and storage artifacts.

Traditional techniques for rapid online measurements of OA without sample handling rely on the combination of mass spectrometry with thermal desorption. Most single particle instruments, e.g.

ATOFMS (Gard et al., 1997), SPLAT (Zelenyuk and Imre, 2005), and PALMS (Murphy et al., 2006), utilize simultaneous laser desorption/ionization, which is not quantitative due to matrix effects and can also result in fragmentation of organic molecules. In contrast, the Aerodyne aerosol mass spectrometer (AMS) utilizes a high vaporization temperature (600 C) and electron ionization (EI, 70 eV) to remain quantitative, but at the cost of extensive thermal decomposition and

ionization-induced fragmentation (Canagaratna et al., 2007). More recently, the CHARON-PTR couples an aerodynamic particle lens with a heated inlet and a softer ionization scheme (via proton transfer reaction) for measurements of particles between 100 and 750 nm (Eichler et al., 2015). This provides improved chemical speciation for some atmospherically-relevant compounds; e.g. oleic acid and 5α-cholestane (Müller et al., 2017). However, proton transfer is too energetic for

studies of the oxygenated compounds characteristic of SOA; for example, only ~10% of *cis*-pinonic acid is detected as the parent ion $[M]H^+$, with the rest distributed across several fragments. Extensive fragmentation in the CHARON-PTR also occurs for oxygenated primary compounds such as levoglucosan limiting its usefulness in mechanistic studies. Another recent development, the AeroFAPA-MS (aerosol flowing atmospheric-pressure afterglow mass spectrometer)

(Brüggemann et al., 2015; Brüggemann et al., 2017) couples thermal vaporization with ionization by the outflow of a low-temperature plasma. The AeroFAPA-MS has detection limits suitable for ambient aerosol measurements and when detecting ions in negative mode is subject to significantly less fragmentation than is PTR, but due to the variety of ions produced in the plasma is subject to various competing ionization pathways, complicating spectral interpretation and quantitative

analysis.

The need for molecular-level information (without ionization-induced fragmentation) led to the development of a number of semi-continuous online measurements that follow a general two-step collection and analysis procedure. Aerosol is typically collected for 10-60 minutes either by impaction or on a filter, then thermally desorbed for gas chromatography with prior online derivatization (Isaacman et al., 2014), proton transfer reaction mass spectrometry (PTR-MS) (Holzinger et al., 2010), or chemical ionization mass spectrometry (CIMS) (Lopez-Hilfiker et al., 2014). These techniques offer significantly improved chemical resolution relative to online systems, but remain subject to thermal decomposition during desorption (though to a lesser degree than the AMS), as well as reaction or partitioning effects on the collection substrate. In some systems which feature a temperature ramp, thermal decomposition products can be identified as such, although links to the parent molecules are unclear (Lopez-Hilfiker et al., 2015). Further, there remain limits to the detection of highly oxidized compounds, as well as accretion products for which there is currently no satisfactory online detection technique. Finally, the time delay between aerosol collection and detection can compromise detection of fast intra-particle reactions (Pospisilova et al., submitted). Therefore, there remains a need for fast, online aerosol analysis without decomposition or fragmentation.

Electrospray ionization (ESI) is a well-known method for transferring low volatility, high molecular weight molecules (e.g. proteins and peptides) into gas phase ions without the need for direct heating. ESI has been successfully coupled to mass spectrometry for offline analysis of atmospheric aerosol both by direct infusion of aerosol extracts (Reemstsma et al., 2006; Zhang et al., 2011) and surface sampling techniques (Laskin et al., 2010; Roach et al., 2010). Extractive electrospray ionization (EESI) is a technique that couples the advantages of ESI with online continuous measurement (Chen et al., 2006). In extractive electrospray ionization, a solvent is delivered through a conventional electrospray probe generating a plume of charged electrospray droplets. The primary spray is directed into a sample flow of gases and/or aerosol whereby the collision between aerosol and electrospray droplets results in the extraction of the soluble components into the bulk electrospray droplet. During the rapid evaporation of the solvent from the electrospray droplets, surface charge is concentrated and ions are ejected into the gas phase, presumably by the Coulomb explosion mechanism (Kebarle and Peschke, 2000). These ions are then sampled directly into a mass spectrometer for analysis.

There have been several previous attempts to apply extractive electrospray ionization to atmospheric gas and particle studies. However, the detection limits achieved by these techniques are one or more orders of magnitude too high to be useful in the atmosphere or for laboratory experiments at atmospherically-relevant concentrations (i.e. ~1-10 µg/m$^3$). Doezema et al. (2012) identified a limited number of compounds in α-pinene SOA at aerosol mass loadings of 1500-2500 µg m$^{-3}$. Ambient ESI, a conceptually similar technique to EESI, detected particle-phase organic compounds, including some oligomers, at 26 µg m$^{-3}$ of SOA from α-pinene ozonolysis, although instabilities in the mass analyzer precluded quantitative analysis (Horan et al., 2012). A more recent EESI system (Gallimore and Kalberer, 2013; Gallimore et al., 2017) attained individual compound detection limits of ~1 µg m$^{-3}$ for 100 s integration (M. Kalberer, private communication), which were further reduced to as low as 0.25 µg m$^{-3}$ using tandem mass spectrometry (MS$^2$) and collisionally-induced dissociation (CID). While a significant improvement compared to previous work, these detection limits remain insufficient for most atmospheric systems, where even the most abundant compounds in OA are typically present in concentrations of 10-100 ng m$^{-3}$. However, EESI was shown to be linear over several orders of magnitude, independent of particle size or morphology up to 200 nm, and reproducible, highlighting its potential benefits for atmospheric OA analysis with soft ionization and without thermal desorption (Gallimore and Kalberer, 2013).

Here we present the development and characterization of a novel EESI interface coupled to a portable high-resolution time-of-flight mass spectrometer (EESI-TOF). Our design, while conceptually similar to previous EESI work, provides detection limits as low as 1 ng m$^{-3}$ at 1 Hz through a combination of source optimization and efficient ion transfer into and through the mass spectrometer. We present comprehensive characterization of the source, including sensitivity, linearity, time response, as well as water vapor and matrix sampling effects. We also present proof of concept measurements from the reaction of α-pinene and ozone in a laboratory flow tube, ambient measurements in Zurich, Switzerland, and airborne wildfire measurements in the central United States.

## 2. Instrument description

In brief, the EESI-TOF system consists of a custom built EESI inlet and ionization source coupled to a commercially available mass spectrometer (APi-TOF, Tofwerk, Thun, Switzerland). In addition to the physical inlet design, selection of working fluid and ionization scheme are critical for optimal EESI-TOF operation.

## 2.1. EESI design

The EESI source was developed with the specific goal to measure OA at a near-molecular level (with "near-molecular" here defined as the determination of a molecular formula, without direct structural information or isomeric separation), however it can more generally be applied to gas or combined gas/particle measurements. Figure 1 shows a schematic of the main components of the EESI. For particle measurements, the instrument automatically alternates between direct sampling and sampling through a Teflon filter with a 1 µm pore size (Fig. 1, orange). The difference between the direct and filter blank measurements is the background-corrected particle-phase signal. For the proof-of-concept deployments discussed in Section 4, this filter was replaced with a nylon cartridge filter (9933-11-NQ, Parker Balston, Lancaster, NY, USA), which performed similarly. Particles and gases enter the EESI source through a 6 mm inner diameter (ID) 5 cm long multi-channel extruded carbon denuder (Fig. 1, black) at a flow rate of 0.7 to 1.0 L min$^{-1}$. The denuder is housed in a stainless steel tube that can be biased relative to the entrance of the mass spectrometer, but is typically set to ground. The denuder improves instrument detection limits by reducing the gas-phase background. In addition, it removes from the particle flow any sticky gases that may be lost to the filter and otherwise misclassified as particulate material.

The denuder is very similar to that used for the CHARON-PTRMS, where it removes methanol, acetonitrile, acetaldehyde, acetone, isoprene, methylethylketone, benzene, toluene, xylene, 1,3,5-trimethylbenzene, and α-pinene with >99.9999% efficiency (Eichler et al., 2015). Our experiments show >99.95% removal for pinonic acid, and no detectable breakthrough in the chamber and field experiments presented in Section 4, which were conducted at OA concentrations up to approximately 10 µg m$^{-3}$ with the denuder not requiring regeneration for at least 2 weeks. For smog chamber experiments on wood and coal-burning emissions at higher concentrations (20 to 200 µg m$^{-3}$, 1 experiment of 3-4 h per day) (Bertrand et al., submitted), the denuder was regenerated every 2 to 3 days, when a slower response time was observed on switching between

the direct sampling and filter blank measurements. This suggests a higher capacity denuder should be used for continuous sampling under polluted conditions.

After the sample passes through the denuder, particles collide with the primary electrospray droplets in a laminar sample flow and the soluble components are extracted. The electrospray is generated by a commercially available 360 μm OD untreated fused silica capillary with an inner diameter of 50 μm (BGB Analytik AG, Boeckten, Germany), with no further treatment. The ESI probe is positioned approximately 1 cm away from the mass spectrometer inlet. A high voltage liquid junction and electrospray fluid reservoir (80 mL) is used to deliver the solvent to the ESI capillary. We find that fluid flow rates between 0.1-10.0 μL/min leads to the formation of a stable primary spray without requiring a sheath gas. Initially, the ESI working fluid was delivered using a commercially available syringe pump (KD Scientific, Holliston, MA, USA) via a 250 μL syringe. However, a combination of unstable fluid delivery rates (pulsed flow) and limited run time (i.e. several hours) led to the replacement of the syringe pump with a high precision pressure regulator for fluid flow (MFCS-EZ 1000 mbar, Fluigent, Inc., Lowell, MA, USA) coupled to a large liquid reservoir. This system allows continuous solvent delivery for months at a time and ensures ultra-stable fluid flow rates.

The droplet-laden flow enters the mass spectrometer through a heated stainless steel capillary to evaporate excess electrospray solvent and facilitate efficient ion formation (Figure 1, blue). The inlet flow rate is determined by the capillary temperature and as noted above is between 0.7 and 1.0 L min$^{-1}$. We use a commercially available stainless steel capillary (0.5 mm inner diameter (ID), 1/16" OD 70 mm long; VICI AG International, Schenkon, Switzerland), which is housed in a conical capillary heater manifold fabricated from aluminum. A tight fit between the heating manifold and capillary helps to ensure uniform heating and efficient heat-transfer to the capillary from the heater block controlled by two cartridge heaters. The heaters are operated at ~250 °C to ensure that electrospray droplets evaporate during the ~1 ms transit through the capillary tube. Note that the combined effects of gas expansion, solvent evaporation, capillary temperature, finite heat transfer (from both cartridge heater to capillary walls and capillary walls to gas), and short residence time results in a sample gas of significantly lower absolute temperature than the 250 °C heater temperature would suggest. The spray probe and assembly is coupled to the mass spectrometer via a PTFE manifold (Fig. 1, green), which thermally isolates the sample flow from the heated capillary manifold (Fig. 1, blue). An aluminum heat sink plate compresses the EESI

mounting manifold to the sample cone of the capillary inlet and draws heat away from the sample flow using a small fan. In this way, the sample flow remains unheated until after extraction into the ESI droplets, minimizing volatilization of labile particle phase components and thermal decomposition.

Evaporation of the charged droplets yields ions via the Coulomb explosion mechanism. These ions are analyzed by a portable high-resolution time of flight mass spectrometer with an atmospheric pressure interface (APi-TOF, Tofwerk, Thun, Switzerland), which has been described in detail previously (Junninen et al., 2010) but modified with a heated capillary inlet. We find that maximum ion transmission is achieved by maximizing the flow rate into the mass spectrometer,
which for our pumping configuration is nominally 1 L min$^{-1}$ at standard temperature and pressure (STP).

### 2.2. Extraction and ionization

Conceptually, the EESI solution consists of two components: (1) a working fluid for aerosol
extraction and spray formation and (2) a dopant for control of ionization pathways. Note that changing one or both of these components alters the set of detectable compounds, increasing the versatility of the EESI-TOF system. In this initial study, we focus on efficient, stable detection of a broad range of organic compounds, with an emphasis on oxygenated SOA. The working fluid and ionization dopant were optimized for the detection of α-pinene ozonolysis products.

Regardless of the working fluid, the electrospray solution is doped with 100 ppm NaI to promote ionization by sodium ion attachment (Na$^+$), and suppress alternative pathways such as proton transfer ([M]H$^+$), formation of adducts with other ions (e.g. Li$^+$ or K$^+$), charge transfer, and proton abstraction ([M-H]$^-$). Positive ion detection is therefore employed. In principle, any compound may be added to the ESI working fluid to control the ionization process, provided that the ion-
adduct binding energies are sufficiently strong to survive droplet evaporation. (For example, detection of I$^-$ adducts in negative mode has so far been unsuccessful, as the [M]I$^-$ adducts dissociate into [M] and I$^-$, presumably during transit through the heated capillary). However, we find that Na$^+$ ions generate strong enough molecular adducts with a wide range of organic molecules present in atmospheric aerosol, including sugars, acids, alcohols, organic nitrates, and
highly oxidized multifunctional molecules.  In principle, Li$^+$ likely provides stronger adducts and

may thus facilitate quantification and improve detection of weakly bound species (Zhao et al., 2017). However, $Na^+$ also has the advantage of only a single peak (no isotopes), which simplifies peak identification and reduces spectral clutter. The detected molecular classes include nearly all compounds present in secondary organic aerosol, with the important exception of organosulfates, which are typically detected as negative ions in electrospray-based studies (Surratt et al., 2008). This ionization scheme is also not sensitive to non-oxygenated compounds such as alkanes, alkenes, and aromatic hydrocarbons. The detectable species are observed exclusively as adducts with $Na^+$. Indeed, the only molecule in ambient or laboratory aerosol that we have identified as ionizing without $Na^+$ attachment is nicotine, where the ionization instead proceeds via net proton transfer, yielding $[M]H^+$ (Qi et al., 2019; Stefenelli et al., 2019). It may be that other reduced nitrogen species follow a similar pathway.

Additional benefits of the $Na^+$ dopant include provision of (1) an internal measure of spray stability and (2) reference ions for $m/z$ calibration. For the former, we typically monitor the $[NaI]Na^+$ ion ($m/z$ 172.883), as for some experiments it is desirable to set the quadrupole guides to block transmission from low $m/z$ ions (including $Na^+$ and working fluid-related signals) to increase detector lifetime. For $m/z$ calibration, we utilize a series of $[(NaI)_n]Na^+$ clusters, which are well-spaced across the entire $m/z$ of interest for ambient aerosol and also have a strong negative mass defect, reducing interferences with organic analytes.

For the ESI working fluid, mixtures of acetonitrile (ACN) (HPLC grade, $\geq$99.9% purity, Sigma-Aldrich, St. Louis, USA) and methanol (MeOH) (UHPLC-MS LiChrosolv, $\geq$99.9% purity, Sigma-Aldrich, St. Louis, USA) in a variable ratio with ultrapure water (18.2 M$\Omega$ cm, total-organic carbon < 5 ppb) were selected for testing; all solutions included the 100 ppm NaI dopant. The solvent blend was optimized with the goals of (1) maximizing the overall OA detection (extraction + ionization) efficiency and (2) generating a spray that is stable over long timescales. We tested a 1:1 MeOH:$H_2O$ mixture and compared this to ACN:$H_2O$ mixtures, as these are the two most common electrospray solvents used in traditional analysis (HPLC and direct infusion ESI) and should have a high overall extraction efficiency of OA. Both solvent blends formed stable electrospray as determined by the stability of detected ion currents over long timescales (days to weeks) and gave similar sensitivities for products from $\alpha$-pinene SOA.

We found that the MeOH:H$_2$O spray produces significant background peaks throughout the spectrum, presumably due to impurities, which are efficiently ionized by the primary electrospray probe. These background signals increase detection limits and complicate interpretation of blank subtraction during the EESI process. Significant effort to ensure the cleanliness of the primary solution is therefore of critical importance for maintaining low detection limits. We found that using a mixture of ACN:H$_2$O for the primary spray generation reduced these backgrounds by approximately an order of magnitude, leading to a net decrease in detection limits. This reduction could simply be due to a more pure acetonitrile solvent compared to methanol. We observe that the effective binding energy of the acetonitrile adduct with sodium $[(ACN)_n]Na^+$ is significantly stronger than that of methanol (Rodgers and Armentrout, 1999; Pejov, 2002). Irrespective of solvent purity, we expect this stronger binding energy to yield a somewhat cleaner spectrum by suppressing subsequent ionization processes. In our test system (α-pinene SOA), we observe the ACN:H$_2$O working fluid can yield clusters of analyte molecules with acetonitrile (i.e., $[M(ACN)]Na^+$), with abundances on the order of 10% of the parent ion ($[M]Na^+$). This effect remains to be characterized in other chemical systems. Note that the cluster abundance depends on the electric fields in the interface and capillary temperature, both of which can be adjusted. Increasing either will decrease the cluster abundance, however at some point the additional energy required to decluster solvent from the organic adducts will reach the binding energy of the organic sodium adducts, thereby reducing overall sensitivity. Note that the sensitivity vs. declustering tradeoff does not affect all species equally and is of greatest importance for the most weakly bound adducts. Herein we present results using mostly the MeOH:H$_2$O spray, unless otherwise explicitly noted.

Preliminary investigations using an H$_2$O-only working fluid (with NaI dopant) were also conducted. This working fluid is of interest because it yields backgrounds even lower than those of the ACN:H$_2$O mixture. However, in our preliminary tests the primary ion count for H$_2$O is also lower by a factor of ~20 relative to ACN:H$_2$O. Further, unlike the MeOH:H$_2$O and ACN:H$_2$O sprays, sampling of ~30 to 35 µg m$^{-3}$ of aerosol from re-nebulized ambient filter extracts resulted in a 10 to 15% decrease in the primary ion signal from the H$_2$O spray, increasing the possibility of ion suppression artifacts or other non-linear behavior. Therefore, while the H$_2$O spray may be of interest for background-limited applications, it cannot be assumed to perform similarly to the MeOH:H$_2$O and ACN:H$_2$O sprays, and detailed characterization is needed.

### 3. Performance and characterization

The EESI-TOF performance was assessed using a variety of single components and atmospherically relevant multi-component aerosol. We focus on assessing sensitivity and detection limits, linearity of response to aerosol mass, and the effects of changes in the OA matrix or water vapor concentrations on EESI-TOF performance.

### *3.1. Test aerosol generation and basic operation*

To characterize and optimize EESI-TOF performance we used both single-component aerosol generated by a conventional nebulizer system as well as multi-component aerosol produced from the reaction of α-pinene and $O_3$ in a flowtube, the configuration of which is described in detail elsewhere (Molteni et al., 2018). The α-pinene is delivered by a diffusion vial from a pure liquid at room temperature into a carrier gas flow of 1-10 L min$^{-1}$ of zero air at the entrance of the flow tube. $O_3$ (0.25-5 ppmv) is produced from a commercially available ozone generator and is mixed into the main flow at the flowtube entrance. This leads to the prompt formation of SOA during the (~1-5 min) residence time in the flow tube. While SOA generated under such conditions is likely not entirely representative of real atmospheric conditions, it contains a suite of highly oxygenated organic monomers, dimers, and higher-order oligomers and therefore provides a useful test aerosol matrix beyond what can be interrogated using a single compound. In this way, we are able to provide a more comprehensive characterization of instrument performance.

Figure 2 shows a sample time series of $[C_{10}H_{16}O_8]Na^+$ measured in SOA generated from α-pinene ozonolysis, with an expanded view shown in the lower panel. Over this measurement period, maximum SOA concentrations reach approximately 30 µg m$^{-3}$. Here the EESI-TOF alternates between 3 min of direct sampling and a 30 s filter blank, denoted by red circles. The background concentration measured during the filter blank is a small fraction of the total signal and is stable over time, typical of our experience for laboratory and atmospheric concentrations of up to at least 100 µg m$^{-3}$. The system rapidly responds to filter actuation, with the signal equilibrating on the order of 5 s. Interposing the filter into the sampled flow causes a small pressure drop, which may slightly perturb the spray; deviations in the $[NaI]Na^+$ signal of up to 2% are typical.

Fundamentally, the EESI-TOF measurement is in terms of the ion flux reaching the detector (Hz), as shown on the left axis of Fig. 2. However, in most studies the particle phase is described in

terms of mass for both absolute and relative concentrations, making it desirable to also obtain a mass-related metric from the EESI-TOF measurements. In principle, the EESI-TOF ion signal for a molecule $x$ can be converted to a mass concentration according to Eq. (1):

$$Mass_x = I_x \left( \frac{MW_x}{EE_x * CE_x * IE_x * TE_{m/z}} \right) * \frac{1}{F} \qquad (1)$$

Here $Mass_x$ denotes the ambient mass concentration of molecule $x$, $I_x$ is the measured ion flux, $F$ is the inlet flowrate (0.7 to 1.0 L min$^{-1}$, depending on inlet capillary temperature), and $MW_x$ is the molecular weight of $x$. Note that $MW_x$ does not generally correspond to the $m/z$ at which $x$ is measured, which is typically an Na$^+$ adduct of $x$. The remaining terms address the probability that a molecule exposed to the electrospray is detected as an ion. The probability that a molecule

dissolves in the spray is defined as the extraction efficiency ($EE_x$). The probability that the analyte-laden droplet enters the inlet capillary is defined as the collection efficiency ($CE_x$). Ions are generated as the droplets evaporate; the probability that an ion forms and survives declustering forces induced by evaporation and electric fields is defined as ionization efficiency ($IE_x$). Finally, the probability that a generated ion is transmitted to the detector is defined as transmission

efficiency ($TE_{m/z}$) and is independent of chemical identity, depending only on $m/z$. We cannot at present distinguish between effects of the four efficiency terms, and so define their product as an empirically-determined compound-dependent response factor ($RF_x$), such that:

$$Mass_x = I_x * \frac{MW_x}{RF_x} * \frac{1}{F} \qquad (2)$$

The $RF_x$ parameter denotes the total number of ions detected per molecule incident to the spray

(i.e. probability that a sampled molecule is detected). Assuming no fragmentation or decomposition, $RF_x$ may be equivalently treated in terms of mass. At present, $RF_x$ has been measured only for a few compounds, and we do not have a reliable parameterization for the many unknowns sampled in laboratory and atmospheric aerosol. Nevertheless, we can arrive at a closer approximation of sampled mass by applying $MW_x$ to calculate the mass flux of $x$ to the detector

($MF_x$):

$$MF_x = I_x * MW_x \qquad (3)$$

The quantity $MF_x$ is used herein for assessment of bulk properties (e.g. comparison of total EESI-TOF signal to external mass measurements and investigation of relative composition). For reference, we show on the right axis of Fig. 3 the $MF_x$ (in attograms ($10^{-18}$ g) per second, ag s$^{-1}$)

corresponding to the measured $I_x$; however, for the remainder of the basic characterization experiments presented herein we show instead the directly measured $I_x$, which is the actual quantity measured by the instrument.

### 3.2. Linearity and sensitivity

To assess EESI-TOF linearity, single-component aerosols were nebulized and sampled simultaneously by the EESI-TOF and a scanning mobility particle sizer (model 3080 differential mobility analyzer and model 3022 condensation particle counter, TSI, Inc., Shoreview, MN, USA). Figure 3 shows the background-subtracted EESI-TOF signal as a function of calibrant mass for two model compounds: raffinose ($C_{18}H_{32}O_{16}$, a surrogate for α-pinene dimers) and dipentaeryrithritol ($C_{10}H_{22}O_7$, a surrogate for isoprene accretion products). Concentrations vary from approximately 1 to 1000 ng m$^{-3}$, thereby covering an atmospherically relevant range of concentrations for single components. In both cases, the molecular species is detected exclusively as adducts of the original molecule with Na$^+$, and both compounds exhibit a linear response to mass in agreement with previous work (Gallimore and Kalberer, 2013). Critically, individual component concentrations of only 10 ng m$^{-3}$ are readily detectable by the EESI-TOF (5 s average). Detection limits based on 3-$\sigma$ variation of adjacent filter blank measurements (i.e. the instrument and spray background) are on the order of a few ng m$^{-3}$. These detection limits improve on the most sensitive previously reported EESI-based aerosol systems by approximately 2 orders of magnitude (or 2-3 orders of magnitude for non-MS$^2$ systems) (Doezema et al., 2012; Horan et al., 2012; Gallimore and Kalberer, 2013) and are sufficient to allow for the first time the detection of OA components in real atmospheric aerosol. Note that these improved detection limits represent the performance of the entire EESI-TOF instrument relative to previous instruments, and we cannot disentangle the contributions of the ionization unit and MS detector.

The different slopes observed between raffinose and dipentaerythritol (4.49 vs. 108 Hz / μg m$^{-3}$) correspond to response factors of $RF_{raff} = 1.88 \times 10^{-8}$ ions molec$^{-1}$ and $RF_{dpe} = 2.93 \times 10^{-7}$ ions molec$^{-1}$, assuming spherical particles with the material density of the pure component. This implies significant differences in the relative sensitivity of the EESI-TOF to different compounds, although as shown later in Fig. 4 dipentaerythritol is an extreme case. Differences are in $RF_x$ expected, however, and may arise from thermodynamic and/or kinetic limitations on extraction efficiency,

as well as ion-adduct binding energies. (In principle, ion suppression, matrix effects, and multiple ionization pathways can also affect sensitivity, though as discussed above and in the next section we do not believe that these issues significantly affect the EESI-TOF). In Fig. 4, we compare the $RF_x$ of a suite of saccharides, polyols, and carboxylic acids, as well as bulk SOA generated by reaction of precursor VOCs with OH radicals in a potential aerosol mass (PAM) flow reactor (Lambe et al., 2011). Because the absolute sensitivity of the EESI-TOF depends on instrument setup (spray optimization, mass spectrometer tuning, etc.), we define a relative response factor ($RRF_x$), using sucrose as a reference:

$$RRF_x = \frac{RF_x}{RF_{sucrose}} \tag{4}$$

Sucrose is chosen as a reference due to its ease of use (i.e. low volatility and high water solubility) and because it is the standard which we have measured most frequently. The pure component $RF_x$ are calculated from SMPS distributions assuming spherical particles with the material density of the pure component, while the SOA $RF_x$ are calculated from the total organic mass of a co-located AMS. Figure 4 primarily shows $RF_x$ determined with the MeOH:$H_2O$ system, although a few ACN:$H_2O$ measurements are included as well.

Several features are evident from Fig. 4. There is a strong decrease in saccharide sensitivity with increasing molecular weight, with the sensitivity of glucose (6 carbon atoms) being nearly 10 times greater than that of glycogen (24 carbon atoms). The relative sensitivities of carboxylic acids and polyols each also span approximately an order of magnitude, although for these classes a clear molecular weight dependence is not observed. The measured polyols also appear to have somewhat higher sensitivities than the other molecular classes, although this feature should be interpreted with caution due to the small number of compounds tested.

The measured SOA mostly follows a trend of decreasing sensitivity with decreasing molecular weight of the precursors, although the high sensitivity of 1,3,5-trimethylbenzene makes it a slight outlier. Calculation of these sensitivities requires the assumption of SOA molecular weights, which were estimated from the EESI-TOF mass spectrum to be 181 g mol$^{-1}$ (benzene), 173 g mol$^{-1}$ (phenol), 195 g mol$^{-1}$ (toluene), 209 g mol$^{-1}$ (naphthalene), 199 g mol$^{-1}$ (α-pinene), and 220 g mol$^{-1}$ (1,3,5-trimethylbenzene). The $RRF_x$ observed for the SOAs span a much smaller range than do the pure components, i.e. a factor 15 between benzene and 1,3,5-trimethylbenzene compared to a factor of ~30 between citric acid and dipentaerythritol (note that the range of $RRF_x$ for pure

components is an underestimate, as some pure components must have an $RRF$ at least as low as benzene, which is itself a factor of 3 lower than citric acid). The smaller $RRF_x$ range exhibited by the SOA experiments is expected given that each value represents the mean $RRF$ of a complex mixture and is consistent with ambient observations, where we do not observe major composition-dependent variations in overall EESI-TOF sensitivity to bulk ambient OA (Qi et al., 2019; Stefenelli et al., 2019). However, direct calibration is clearly advisable for compounds for which absolute quantification is desired. Further, for aerosol of unknown composition the possibility of multiple isomers (potentially having significantly different $RRF_x$) must be considered.

The SOA species shown in Fig. 4 are comprised of many individual compounds, and it is highly desirable to constrain their relative concentrations and thus $RRF_x$. However, direct calibration of every compound is not feasible due to the large number of species present; in addition, many SOA compounds are not commercially available and cannot be readily synthesized. Therefore, to better understand the relative sensitivities of the individual ions, we utilize as a reference the well-characterized FIGAERO-I-CIMS (filter inlet for gases and aerosols, coupled to iodide chemical ionization mass spectrometry) (Lopez-Hilfiker et al., 2014). The FIGAERO collects particles for 30 min, after which a 45 min thermal desorption program is applied and the resulting organic vapor is detected by I-CIMS. Reaction rates in the FIGAERO-I-CIMS are collision-limited, which in conjunction with ion-adduct binding energies and operational characterization of the declustering potential within the ion transfer optics allows estimation of the sensitivity of the instrument to compounds for which standards do not exist. A direct compound-to-compound comparison between the EESI-TOF and FIGAERO-I-CIMS thus allows us to assess how $RRF_x$ obtained by the EESI-TOF compare to an ionization/detection scheme that can be well-described theoretically in terms of fundamental principles (Iyer et al., 2016; Lopez-Hilfiker et al., 2016).

As a test aerosol, we again select SOA from α-pinene ozonolysis, for which many of the product compounds are detected at or near the collision limit by the FIGAERO-I-CIMS. Although this comparison is shown on a logarithmic scale due to the range of signal intensities recorded, strong linear correlations are observed for every ion. Moreover, a single line can reasonably describe the EESI-TOF vs. FIGAERO-I-CIMS correlation, regardless of ion identity. This is further highlighted in Fig. 5c, where the slopes of every ion from the comparison in Fig. 5b are shown; the spread of slopes describes the range of relative sensitivities in the EESI-TOF relative to the FIGAERO-I-CIMS. The mean slope is 0.51, with a standard deviation of 0.11, indicating that the

instrument-to-instrument sensitivity typically varies by only ±20%, and all ions are within ±40% of the mean. Also of note is a slight reduction in the relative sensitivity of the EESI-TOF to the FIGAERO-I-CIMS for the smaller $[C_9H_{14}O_x]Na^+$ series (0.45 ± 0.09) compared to $[C_{10}H_{16}O_x]Na^+$ (0.57 ± 0.08); however, additional measurements are needed to validate this trend. In all, the strong correlation of the EESI-TOF with the collision-limited FIGAERO-I-CIMS suggests that the FIGAERO-I-CIMS strategy of drawing on a fundamental limit to parameterize sensitivity to unknown compounds may likewise be applicable to the EESI-TOF.

### 3.3. Matrix effects

Matrix effects and ion suppression processes are common in direct infusion electrospray sources, introducing non-linear responses to analyte concentration and impeding quantification efforts (Stüber and Reemtsma, 2004; Cappiello et al., 2008; Furey et al., 2013). We therefore characterized EESI-TOF sensitivity to a test compound (dipentaerythritol) in the presence of a complex and variable particle-phase organic matrix produced from α-pinene ozonolysis, used again here as a surrogate for the multi-component aerosol particles present in the atmosphere. Dipentaerythritol was chosen as a reference because it does not evaporate during transit through the flow tube, is relatively unreactive, and has a chemical formula ($C_{10}H_{22}O_7$) not found in α-pinene ozonolysis products (maximum of 18 hydrogen atoms for a $C_{10}$ molecule). A constant concentration of pure dipentaerythritol particles was introduced into the flow tube, and coated with SOA formed from α-pinene ozonolysis, generated as described above. Coating thickness was controlled by varying the $O_3$ concentration. The initial dipentaerythritol particles are approximately 20 nm in diameter and increase to ~70 nm by coating with α-pinene SOA, corresponding to approximately 75 μg m$^{-3}$ of SOA. At these high SOA concentrations, the α-pinene ozonolysis reaction produces sufficient concentrations of low-volatility organics to induce nucleation, which competes with the dipentaerythritol test particles as a surface for condensing SOA. At the maximum SOA concentration, analysis of the SMPS size distributions shows that approximately 40% of the mass corresponds to coated dipentaerythritol, and 60% to nucleated α-pinene SOA.

A change in the measured concentration of dipentaerythritol in response to the SOA coating would indicate a matrix-dependent instrument response. Figure 6 shows the measured dipentaerythritol

signal ($[C_{10}H_{22}O_7]Na^+$) as a function of $[C_{10}H_{16}O_7]Na^+$, a major ion in α-pinene SOA that is approximately proportional to the total SOA mass. Measurements are colored by time. Even at the thickest coatings, the depentaerythritol signal is unaffected, indicating that particle-phase matrix effects do not significantly affect the measurement. Although these results are obtained from a single test system, they are consistent with the general trend of matrix effects in EESI studies that are negligible or strongly suppressed relative to conventional electrospray measurements (Chen et al., 2006; Gu et al., 2007; Zhou et al., 2007; Chen et al., 2009). In addition, we find that OA signals measured by the EESI-TOF are well correlated with AMS measurements (e.g. total measurable OA, source apportionment factors, tracer ions) (Qi et al., 2019; Stefenelli et al., 2019). This suggests that EESI-TOF bulk OA measurements are likely not affected by soluble inorganic matrices typical of Central Europe (i.e., internal mixtures with $NH_4NO_3$ and $(NH_4)_2SO_4$ concentrations up to ~10 µg m$^{-3}$), although effects on individual ions cannot be ruled out.

### 3.4. Water vapor dependence

Atmospheric water vapor concentrations are high in absolute terms and highly variable, and as a result affect the response of instruments based on chemical ionization (Vlasenko et al., 2010; Lee et al., 2014; Iyer et al., 2016; Zhao et al., 2017). Water can potentially decrease sensitivity by competing with the analyte for $Na^+$ ions (e.g. by displacing the analyte), or increase sensitivity by absorbing energy from $Na^+$-adducts, thereby stabilizing them. We therefore investigated the water vapor-dependent response of the α-pinene ozonolysis SOA mass spectrum. For these tests, SOA was generated at constant concentration in a flow tube and programmatically diluted by a dry and a wet flow. The wet flow was humidified by passing through a water bubbler held at room temperature (~25 °C). The total dilution was kept constant, but the ratio of the wet and dry flows was systematically varied to obtain relative humidities ranging from 0 to 80%. Assuming a sample flow of 0.8 L min$^{-1}$ and 1 to 10 µL min$^{-1}$ flow through the electrospray capillary, the working solution provides 7 to 42 % of the total water at 50 % RH. Water vapor has a significant memory effect in the denuder used in the EESI-TOF for removing semi-volatile gases from the sample flow, therefore despite step changes in the dilution relative humidity the water vapor seen by the instrument changes much more smoothly. As a surrogate for water vapor concentration, we monitor a sodium iodide adduct generated in the spray and determine the ratio of its water-

clustered and water-free forms, i.e. $[NaI(H_2O)]Na^+$ / $[NaI]Na^+$. Note that the particulate water content is insignificant compared to the ESI droplets and sample flow water vapor, and thus does not perturb either the $[NaI(H_2O)]Na^+$ / $[NaI]Na^+$ ratio or the EESI-TOF sensitivity.

The left axis of Fig. 7a (black dots) shows an example of the $[C_{10}H_{16}O_7]Na^+$ ion as a function of the $NaI(H_2O)Na^+$ / $NaINa^+$ ratio. For this ion, the instrument response is constant independent of water vapor concentration, i.e. the ratio of the ion signal ($I$) at a given RH to the ion signal at 0% RH ($I/I_{RH=0}$) is ~1. On the right axis (red line and shading), we show the median, 10th, and 90th percentiles for $I/I_{RH=0}$ across all detected ions as a function of $[NaI(H_2O)]Na^+$ / $[NaI]Na^+$; $I/I_{RH=0}$ = 1 is shown as a dashed green line for reference. Probability distributions of $I/I_{RH=0}$ at $[NaI(H_2O)]Na^+$ / $[NaI]Na^+$ = 0.1, 0.5, and 0.9 (with the latter condition corresponding to ~80% RH) are shown in Fig. 7b. As water vapor increases, the $I/I_{RH=0}$ distribution broadens, and the median signal decreases slightly (~15% lower at 80% RH). Note that the broadening takes place on the low-sensitivity side of the $I/I_{RH=0}$ distribution; Fig. 7b shows that many ions have a humidity-independent response, similar to $[C_{10}H_{16}O_7]Na^+$. Even for ions with the most extreme humidity dependence, at the highest water vapor concentrations measured for 90% of ions $I/I_{RH=0} > 0.60$ and for 99% $I/I_{RH=0} > 0.45$. This weak perturbation by ambient water vapor simplifies spectral interpretation, and suggests that ion chemistry occurs predominantly in the droplet phase.

### 3.5. Effect of particle size

Particle size can in theory affect detection in the EESI-TOF system via two mechanisms. First, particle losses may occur within the denuder by diffusion or impaction, affecting small and large particles, respectively. Second, larger particles may be incompletely extracted in the spray. These two possibilities are investigated separately. Figure 8a shows the denuder transmission efficiency as a function of particle size, as measured by a scanning mobility particle sizer (SMPS). For this test, the denuder is removed from the EESI inlet and placed within a segment of straight tubing. SMPS measurements before and after the denuder are compared to determine transmission efficiency. The figure shows that particles are transmitted with better than 80% efficiency over the measured size range of 20 to 750 nm.

The size dependent response of the EESI extraction/ionization processes was characterized using pure dipentaerythritol particles nebulized from an aqueous solution. The polydisperse particles

were dried and quantified using an SMPS system that sampled in parallel with the EESI-TOF. The size of the nebulized particles was controlled by changing the nebulizer flowrate and solution concentration. Figure 8b shows the measured sensitivity of the EESI-TOF to dipentaerythritol ($C_{10}H_{22}O_7$) as a function of the (polydisperse) particle volume distribution geometric mean diameter, which ranges between approximately 50 and 250 nm. The same sensitivity is measured independent of particle diameter, indicating complete extraction into the droplet phase and a lack of any size-dependent ionization due to solvation kinetics or incomplete extraction. Previous investigations of this effect in EESI-based aerosol systems have yielded conflicting results; our results are consistent with those of Gallimore and Kalberer (Gallimore and Kalberer, 2013), who observed no size dependence for single component OA particles with diameters ≤200 nm. For ambient aerosol, we observe a linear relationship with mass for particles of up to ~500 nm diameter (with larger sizes not yet investigated) for both methanol:water and acetonitrile:water working fluids (Qi et al., 2019; Stefenelli et al., 2019). In contrast, Kumbhani et al. (Kumbhani et al., 2018) observed incomplete extraction of $NaNO_3$ particles coated with glutaric acid, also using 1:1 methanol:water as the working fluid. One possibility is that the characteristics of the generated electrospray are significantly different in the EESI-TOF, e.g. larger droplet diameter, increased droplet number density, and/or longer droplet/analyte contact time.

We expect that the electrospray droplets are on the order of 4 to 40 µm, significantly larger than the sampled aerosol (Smith et al., 2002; Wortmann et al., 2007; Soleilhac et al., 2015; Liigand et al., 2017). In principle, as particles grow to larger diameters the aerosol particle diameter could begin to approach the diameter of the electrospray droplets. In such a limit, the extraction process would become less efficient, as particles may bounce or incompletely dissolve in the electrospray droplet plume (Wang et al., 2012). The electrospray liquid flow rate controls the electrospray droplet size distribution; therefore it is important to ensure that the flow rate is sufficiently high to keep the electrospray droplet diameter much larger than the particles to be analyzed. Low electrospray flow rates increase instrument sensitivity by up to a factor of two (a similar trend is frequently observed for conventional ESI), but decrease response time and can introduce a size-dependent extraction efficiency at larger particle sizes. We hypothesize that these effects are due to slower flushing of the electrospray tip and surrounding areas, as well as a shift of the primary electrospray droplets towards a smaller size distribution, affecting charge density. In addition, we performed some preliminary experiments during the development of the EESI-TOF using drawn

electrospray tips to generate the primary EESI droplets. Although drawn tips are known to be very efficient at generating ions in conventional electrospray sources, we found that with 15-30 μm drawn capillary tips (New Objective, Inc., Woburn, MA, USA) aerosol extraction was not efficient despite higher total primary spray ion currents, presumably due to the electrospray droplet size distribution being much smaller than our standard electrospray capillary.

## 4. Proof of concept measurements and applications

To assess the versatility and robustness of the EESI-TOF, proof-of-concept measurements were conducted across several laboratory and field-based platforms. Here we present sample results from the measurement of SOA generated from α-pinene ozonolysis in an environmental chamber, ground-based ambient measurements, and a proof-of-concept deployment aboard a research aircraft. The EESI-TOF performed successfully in each of these environments, demonstrating its potential for a wide range of measurement applications.

### 4.1. Laboratory chambers

Dark ozonolysis of α-pinene was investigated using the 27 m$^3$ PSI environmental chamber (Paulsen et al., 2005). Briefly, the chamber was filled with 40 to 50 ppb of ozone at 40 % RH, after which 30 ppb of α-pinene was injected and the ensuing reaction proceeded undisturbed under dark conditions for approximately 16 hours. The EESI-TOF monitored the composition of SOA particles with 1 s time resolution. A detailed analysis of the chemical composition and evolution of SOA from the α-pinene ozonolysis system as determined by the EESI-TOF is provided elsewhere, and only proof-of-concept sample data is shown here (Pospisilova et al., submitted). The time series of the summed signal of all $[C_xH_yO_z]Na^+$ ions recorded by the EESI-TOF and the measured SMPS mass are shown in Fig. 9a, with these two values compared as a scatterplot in Fig. 9b. A strong linear correlation between the EESI-TOF and SMPS signal is observed throughout the experiment.

Figures 9c and 9d show an averaged EESI-TOF mass spectrum collected from the period of maximum suspended aerosol mass. The raw (background-subtracted) spectrum is shown in Fig. 9c. To aid the eye, $[(NaI)_n]Na^+$ clusters are removed; this is done because although the background-subtracted $[(NaI)_n]Na^+$ is close to zero, it is the difference of two high-intensity signals and therefore remains large relative to most ions in the mass spectrum. Distinct regions are clearly

visible for monomers, dimers, and higher-order oligomeric species. The dimer region is shown (on a linear axis) in the inset. Figure 9d summarizes the chemical composition in terms of a mass defect plot (difference between ion exact mass and the nearest integer, as a function of integer mass) for ions having between 7 and 10 carbon atoms. The ions are colored by carbon number and the points are sized by the signal intensity. This highlights the detailed chemical information provided by the EESI-TOF, as well providing an example of the compositional trends running throughout the spectrum.

### 4.2. Ambient ground-based measurements

The EESI-TOF was deployed at an urban site in Zurich, Switzerland for approximately 3 weeks during summer 2016. Stable operation was achieved for >85% of the sampling period. The EESI-TOF mass spectral time series were used to identify sources and processes governing OA in Zurich, both by direct inspection of chemical signatures and using the positive matrix factorization source apportionment technique (Stefenelli et al., 2019). Figure 10 shows the EESI-TOF mass spectrum averaged over the entire campaign, colored to highlight specific ions and families. In dark blue, we show again the set of the $[C_{7-10}H_xO_y]Na^+$ ions observed in α-pinene SOA in the PSI environmental chamber (see Fig. 9d). These ions are among the strongest contributors to the ambient OA signal during this campaign. These ions are not necessarily unique to α-pinene ozonolysis, as many can also be generated as reaction products of other terpenes and/or different oxidants (e.g. OH, $NO_3$ radicals). A subset of these ions may also derive from ring-opening oxidation products of aromatics. However, in a general sense, they highlight the strong contribution of biogenic emissions to summer OA in Zurich, consistent with previous studies (Daellenbach et al., 2017).

Several other species of note are highlighted in Fig. 10. The single most intense peak in the spectrum ($[C_6H_{10}O_5]Na^+$, $m/z$ 185.042) is attributed to levoglucosan and its isomers. The high intensity occurs because of the significant contribution of levoglucosan to OA from biomass combustion, as well as the high sensitivity of the EESI-TOF to small saccharides (see Fig. 4). Another high intensity ion is attributed to nicotine ($[C_{10}H_{14}N_2]H^+$, $m/z$ 163.123), which as a reduced nitrogen species ionizes by hydrogen abstraction rather than $Na^+$ adduct formation. This introduces considerable uncertainty into both the sensitivity and linearity of the EESI-TOF

response to nicotine. However, the AMS nicotine tracer $C_5H_{10}N^+$ (Struckmeier et al., 2016) exhibits a strong linear correlation with the EESI-TOF nicotine measurement, suggesting that no significant non-linearities are present under the conditions encountered in Zurich (Qi et al., 2019). Finally, the spectrum shows a significant contribution from $[C_xH_yO_zN_{1-2}]Na^+$ ions, which are mostly assigned to organonitrates as discussed in detail elsewhere (Stefenelli et al., 2019).

Estimated detection limits (30 s) are shown on an ion-by-ion basis in Fig. 10b, using the same color scheme as in Fig. 10a. Detection limits are calculated as the 3-$\sigma$ variation of the ion signal during the filter blank periods flanking a direct sampling interval, and the campaign median is shown. Detection limits are converted to mass assuming a uniform sensitivity of 1450 Hz / (μg m$^{-3}$), which corresponds to the estimated sensitivity of SOA during this campaign (Stefenelli et al., submitted). This is a rough estimate which neglects ion-dependent sensitivities and differences in molecular weight. The results are summarized in histogram form in Fig. 10c. Most species have detection limits in the range 1 to 10 ng m$^{-3}$ (median 5.4 ng m$^{-3}$), with nitrogen-containing ions having slightly lower detection limits than other species. We note that these measurements utilized a methanol:H$_2$O working solution, and detection limits from the cleaner acetonitrile:H$_2$O system will likely be lower.

### 4.3. Aircraft deployment

Mobile sampling platforms, such as aircraft and ground vehicles, require highly time-resolved measurements. As discussed above, current OA measurements used in mobile measurements require a tradeoff between thermal decomposition (extensive for the AMS, minor for FIGAERO-CIMS), ionization-induced fragmentation (AMS, CHARON-PTRMS), or time resolution (FIGAERO-CIMS). The EESI-TOF thus addresses an important gap in mobile aerosol instrumentation. As a proof of concept, the EESI-TOF was deployed aboard the NOAA C-130 aircraft during the Airborne Research Instrumentation Testing Opportunity (ARISTO) 2016 test flight campaign from August 1 to 19, 2016. For these flights, the EESI inlet was installed on an API-TOF previously configured for research flights by the University of Washington (Lee et al., 2018) and operated on a pressure-controlled sampling line. In general, good performance and stability were achieved. The main difficulty encountered was icing on the sampling inlet outside the aircraft, which reduced the line pressure and flow and thereby altered the ESI spray.

Figure 11a shows a time series of $[C_6H_{10}O_5]Na^+$, which corresponds to levoglucosan and its isomers. Also shown are $[C_{10}H_{16}O_4]Na^+$ and $[C_{10}H_{16}O_5]Na^+$, which are major products of SOA formed from monoterpene oxidation. EESI-TOF data were collected at 1 Hz; each point in the figure corresponds to a 20 s re-average. At approximately 17:35, the aircraft intersects a wildfire

plume. A dramatic increase in $[C_6H_{10}O_5]Na^+$ is observed, compared to only minor changes in the $[C_{10}H_{16}O_x]Na^+$ ions. The high time resolution of the EESI-TOF is critical for accurate characterization of this plume, as the entire period of intersection lasts only ~3 min. Figure 11b shows a 20 s average mass spectrum, corresponding to the period of maximum $[C_6H_{10}O_5]Na^+$ concentration. The spectrum is normalized to levoglucosan, which is the most intense peak.

However, even for such a short averaging interval, many other ions are evident throughout the mass spectrum, indicating the wealth of chemical information accessible even for highly time-resolved and/or mobile measurements.

## 5. Conclusions

We present an extractive electrospray ionization (EESI) source coupled to TOF-MS for laboratory and field measurement of OA on a near-molecular level. The EESI-TOF achieves detection limits compatible with operation at ambient aerosol concentrations, making it the first instrument capable of real-world OA measurements without thermal decomposition, ionization-induced fragmentation, competitive ionization pathways, or separated collection/analysis stages. We

observe an instrument response that is linear with mass and without a detectable dependence on the composition of the OA matrix for a dipentaerythritol/α-pinene SOA test system. Ambient measurements also suggest that bulk OA detection is not significantly affected by a changing matrix of soluble inorganic compounds. Changing water vapor concentrations only slightly affect the instrument response to most ions, with a ~50% decrease in sensitivity observed for the most

extreme cases. Particle transmission to the EESI source is greater than 80% between 20 and 750 nm, with particles of at least 250 nm (and likely 500 nm) completely extracted in the spray (larger particles remain to be tested). The EESI-TOF was successfully deployed for environmental chamber experiments, ground-based ambient sampling, and tests flights aboard a research aircraft, highlighting its versatility and range of potential applications.

The EESI-TOF sensitivity to SOA generated from a set of individual precursor gases varies within a factor of 15. Larger variations in sensitivity were found between pure organic standards. However, compound-by-compound sensitivities in laboratory-generated SOA are proportional to those determined by the collisionally-limited FIGAERO-I-CIMS within ±50%. This shows promise for an eventual parameterization of the EESI-TOF sensitivity to unknown species, and further suggests that the EESI-TOF responds in a similar way to collision-limited CIMS approaches and that the mass spectra thus approximately reflect the actual distribution of detectable compounds.

The working fluid and ionization scheme plays an important role in both instrument stability/performance and the set of detectable compounds. Here we have focused on positive ion detection using a 1:1 methanol:water system (with a subset of data using 1:1 acetonitrile:water), with 100 ppm NaI added to suppress all ionization pathways except for $Na^+$-adduct formation. This controlled ionization is highly desirable, and contributes to the system linearity, lack of matrix effects, and spectral interpretation. These $Na^+$-based systems allow detection of most compounds comprising atmospheric OA, with the notable exceptions of non-oxygenated species and organosulfates. However, different extraction/ionization schemes could be envisaged for different chemical targets, increasing the overall utility of the EESI-TOF.

**Acknowledgements**

This work was supported by the Swiss National Science Foundation (starting grant BSSGI0_155846). The ARISTO test flight program is funded by the U.S. National Science Foundation Deployment Pool. We thank Tofwerk AG and the PSI machine shop for supporting the construction and integration of the EESI source. We also thank David Bell, Deepika Bhattu, and Yandong Tong (PSI) for useful discussions on acetonitrile clustering, $H_2O$ working fluid, and large particle detection. Rene Richter and Christoph Hueglin are gratefully acknowledged for logistical support of the Zurich deployment.

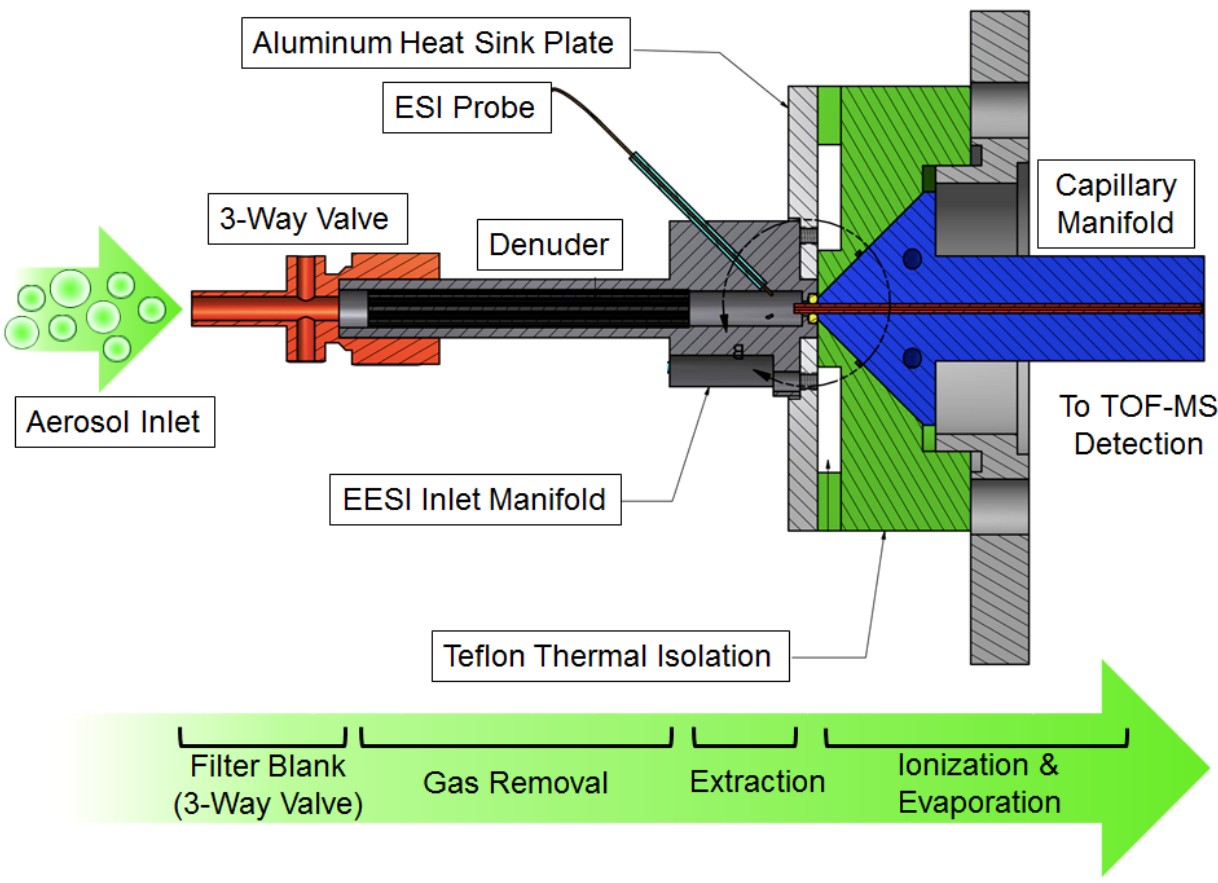

**Figure 1.** Schematic of the EESI-TOF inlet and ion source, including connection to TOF-MS.

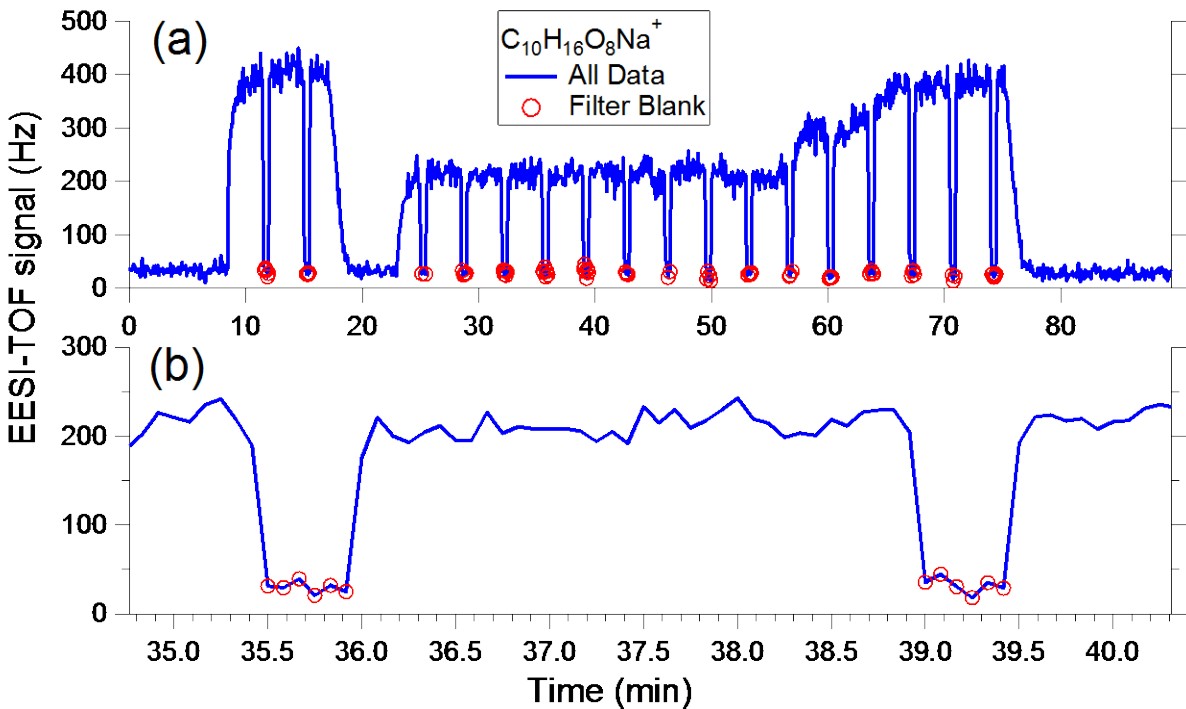

**Figure 2.** (a) Sample time series of $[C_{10}H_{16}O_8]Na^+$ measured in SOA generated from α-pinene ozonolysis, showing aerosol measurement periods (3 min) interspersed with filter blanks (30 s). The difference between these two conditions yields the signal due to sampled particles. (b) Expanded view of two measurement/filter cycles showing; instrument response to filter switching is ~5 s.

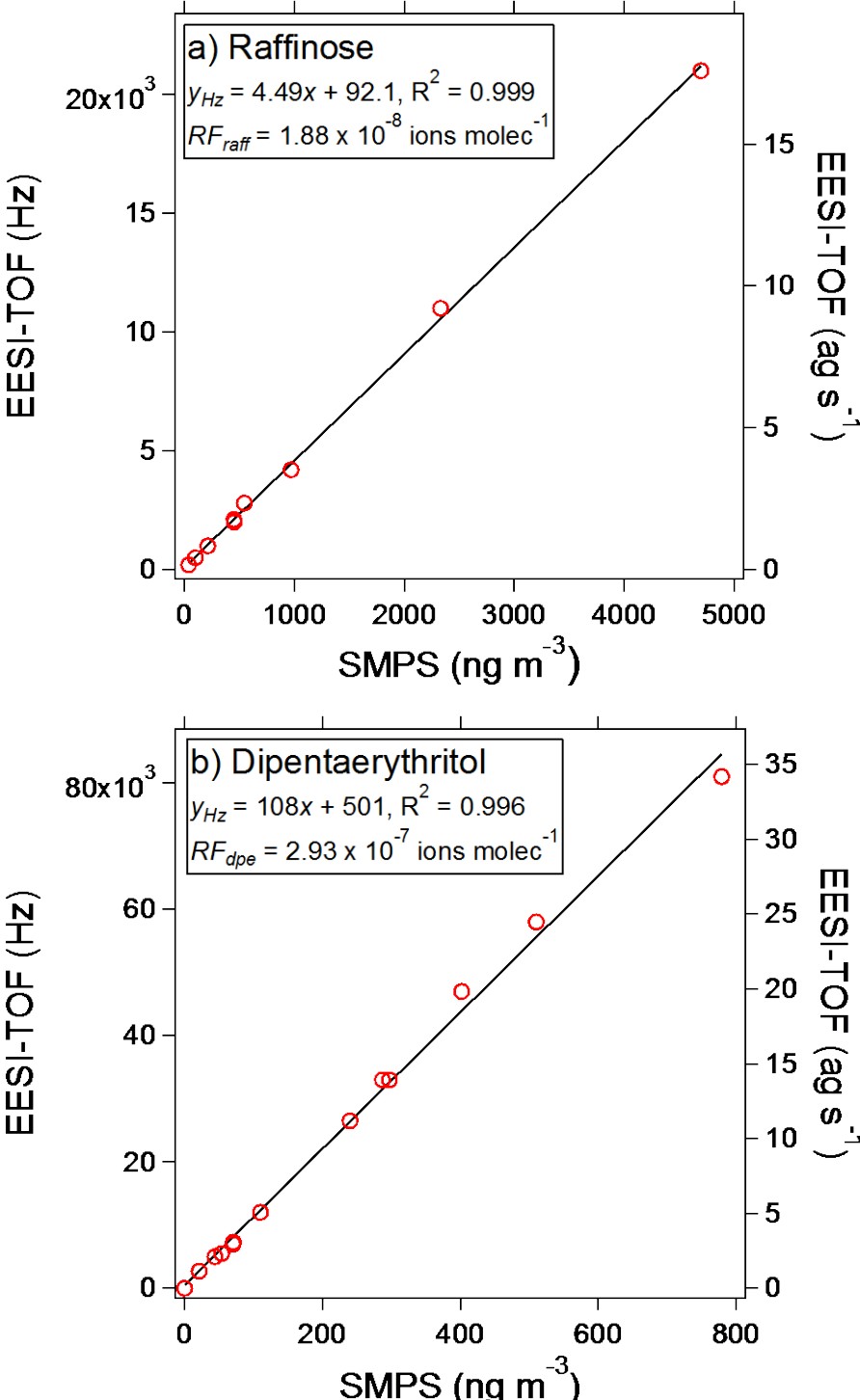

**Figure 3.** EESI-TOF signal as a function of mass concentration measured by an SMPS, assuming spherical particles with the material density of the pure compound, for raffinose (a) and dipentaerythritol (b). The EESI-TOF signal is represented both in terms of the flux of ions (left axis) and mass (right axis) reaching the detector.

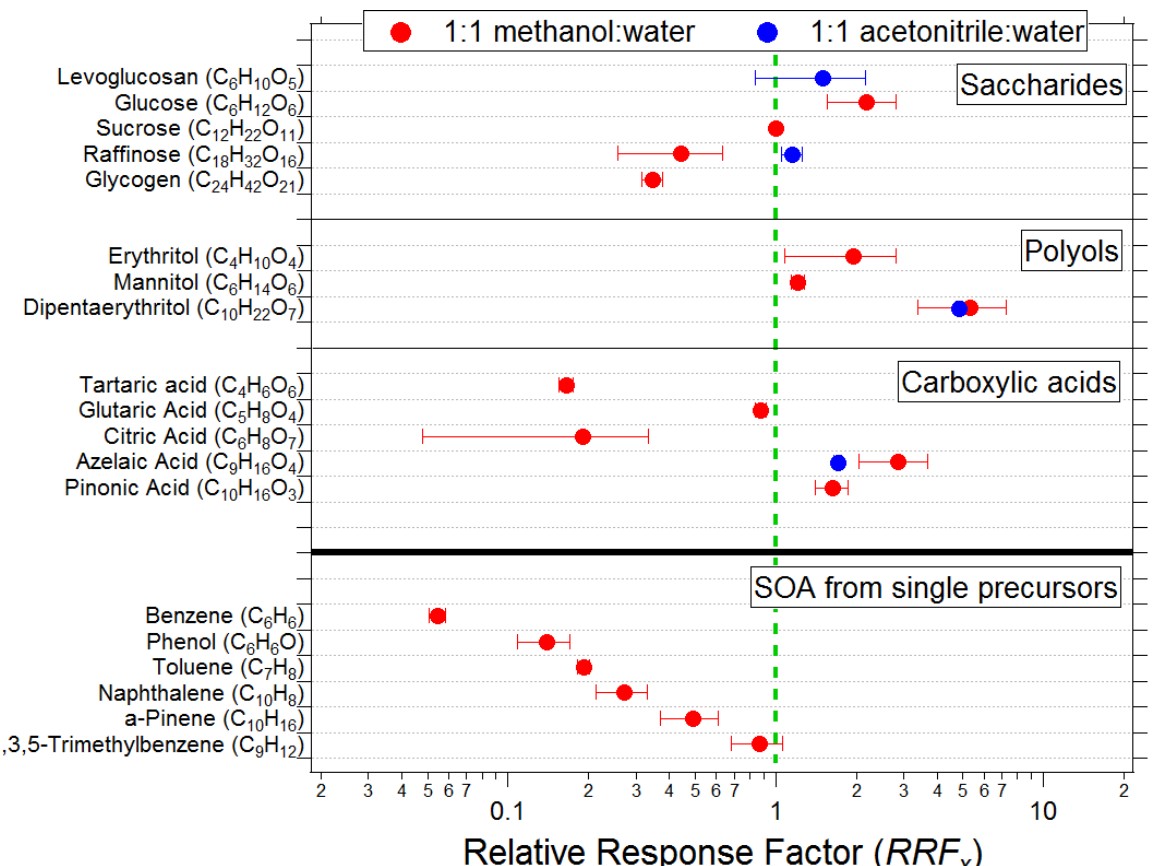

**Figure 4**. EESI-TOF sensitivity (ions molecule$^{-1}$, see Eq. 1) relative to that of sucrose for pure components and SOA formed by OH-initiated oxidation of single precursors in a PAM flow reactor. Red points denote a 1:1 methanol:water working fluid; blue denotes 1:1 acetonitrile:water. All configurations use 100 ppm NaI as a dopant, and all ions are detected as [M]Na$^+$.

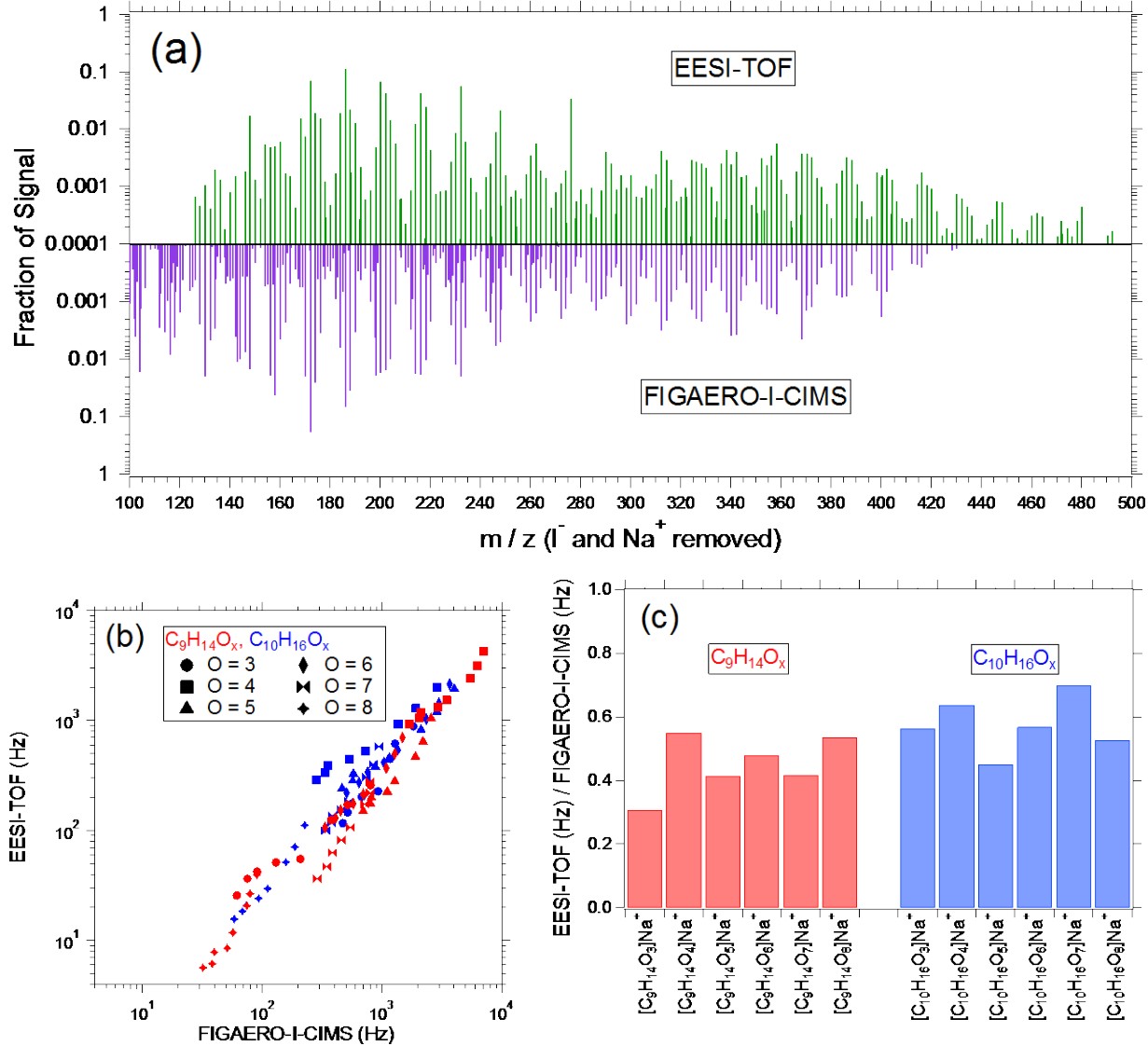

**Figure 5.** (a) Particle-phase mass spectra of SOA from α-pinene ozonolysis measured by the EESI-TOF (top, green) and FIGAERO-I-CIMS (purple, bottom). Spectra are normalized such that the sum across the displayed $m/z$ window is 1, and the reported $m/z$ are after subtraction of $Na^+$ (EESI-TOF) and $I^-$ (FIGAERO). (b) EESI-TOF signal (Hz) as a function of FIGAERO-I-CIMS (Hz) for the $[C_9H_{14}O_x]Na^+$ (red) and $[C_{10}H_{16}O_x]Na^+$ (blue) ion series. Marker shape denotes number of oxygen atoms. (c) Slopes of linear fits to individual ions for the $[C_9H_{14}O_x]Na^+$ (red) and $[C_{10}H_{16}O_x]Na^+$ (blue) ion series. All fits are conducted using orthogonal distance regression with unconstrained intercepts.

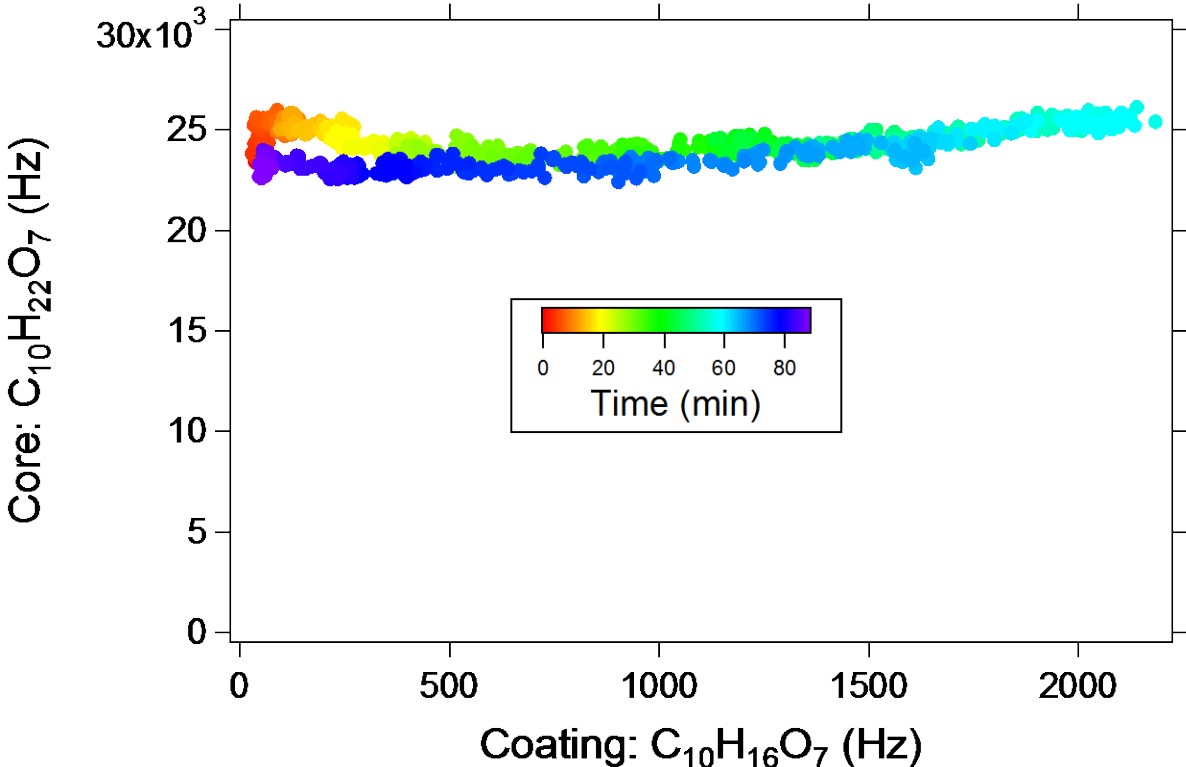

**Figure 6.** Dipentaerythritol $[C_{10}H_{22}O_7]Na^+$ signal measured in a flow tube as a function of $[C_{10}H_{H16}O_7]Na^+$ signal, which is proportional to condensed SOA mass. The maximum $[C_{10}H_{H16}O_7]Na^+$ signal corresponds to approximately 75 μg m$^{-3}$ of SOA, of which approximately 40% occurs as a coating on the dipentaerythritol seed, increasing the particle diameter from 20 to 70 nm. Points are colored by time, showing an increase and then decrease of coating material.

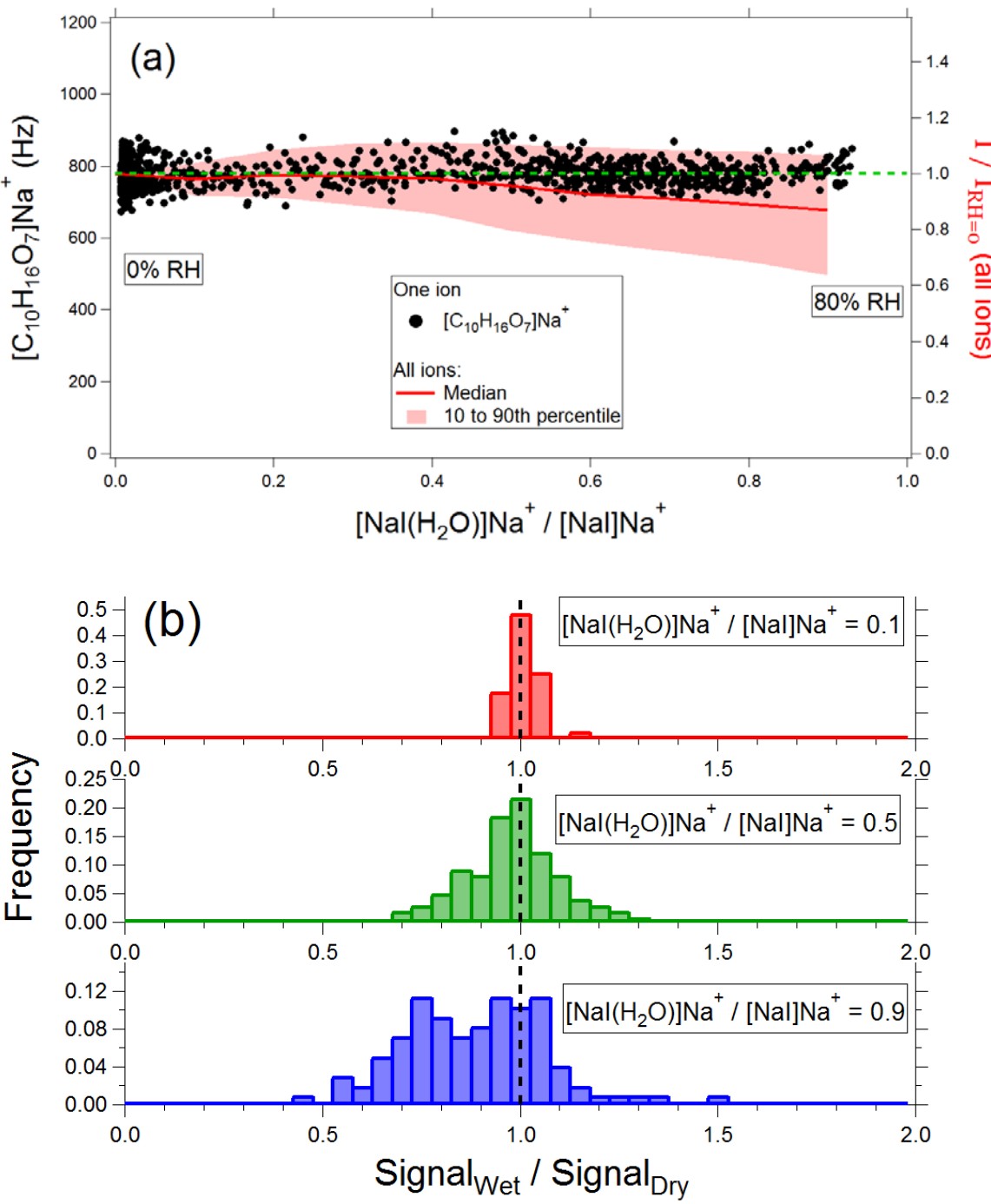

**Figure 7.** Effect of water vapor on EESI-TOF response. (a) Left axis, black points: $[C_{10}H_{16}O_7]Na^+$ signal as a function of $[NaI](H_2O)]Na^+ / [NaI]Na^+$. Right axis, red shading: median, 10th, and 90th percentiles of $I/I_{RH=0}$ for all measured ions. (b) Probability distributions of $I/I_{RH=0}$ at $[NaI](H_2O)]Na^+ / [NaI]Na^+ = 0.1$, 0.5, and 0.9, where $[NaI](H_2O)]Na^+ / [NaI]Na^+ = 0.9$ corresponds to 80% RH.

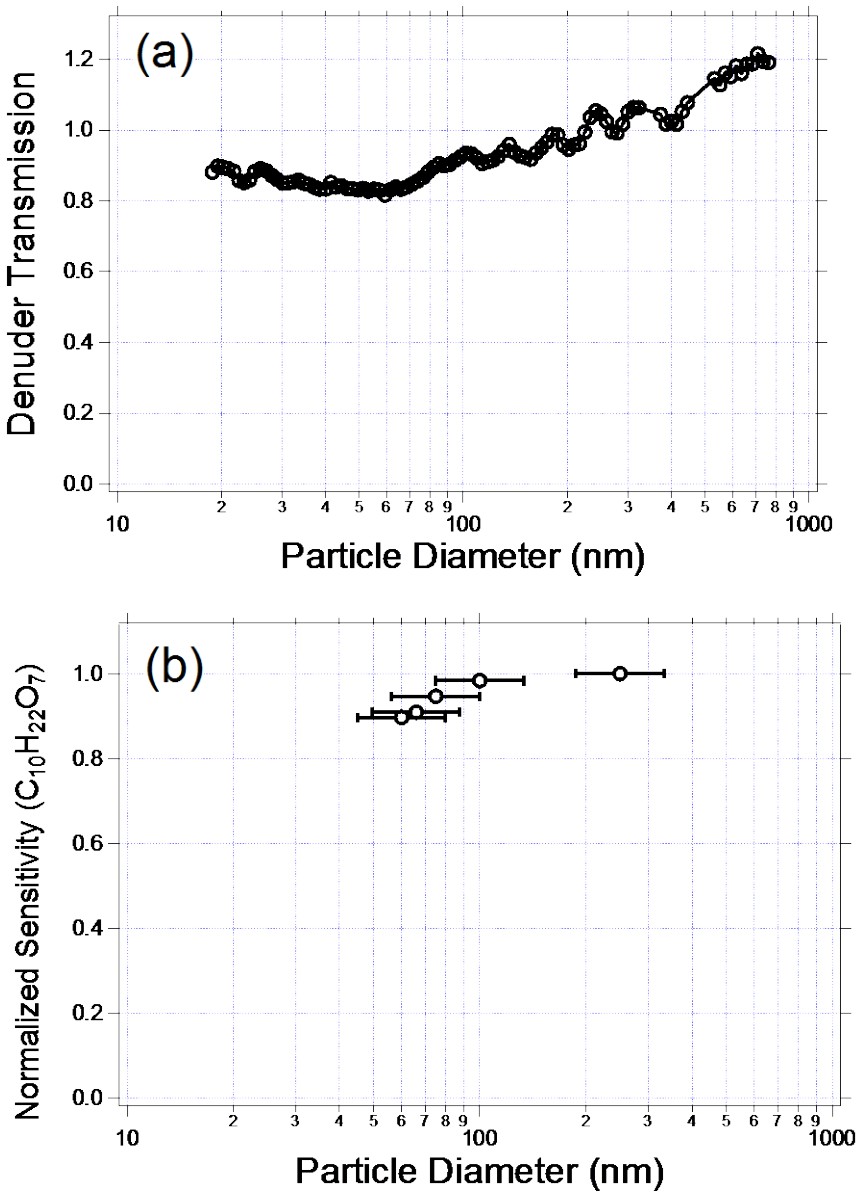

**Figure 8.** (a) Particle transmission through the denuder as a function of diameter. (b) Measured sensitivity of dipentaerythritol as a function of particle diameter. Error bars denote the mobility diameters corresponding to the half-maxima of the polydisperse particle volume distribution.

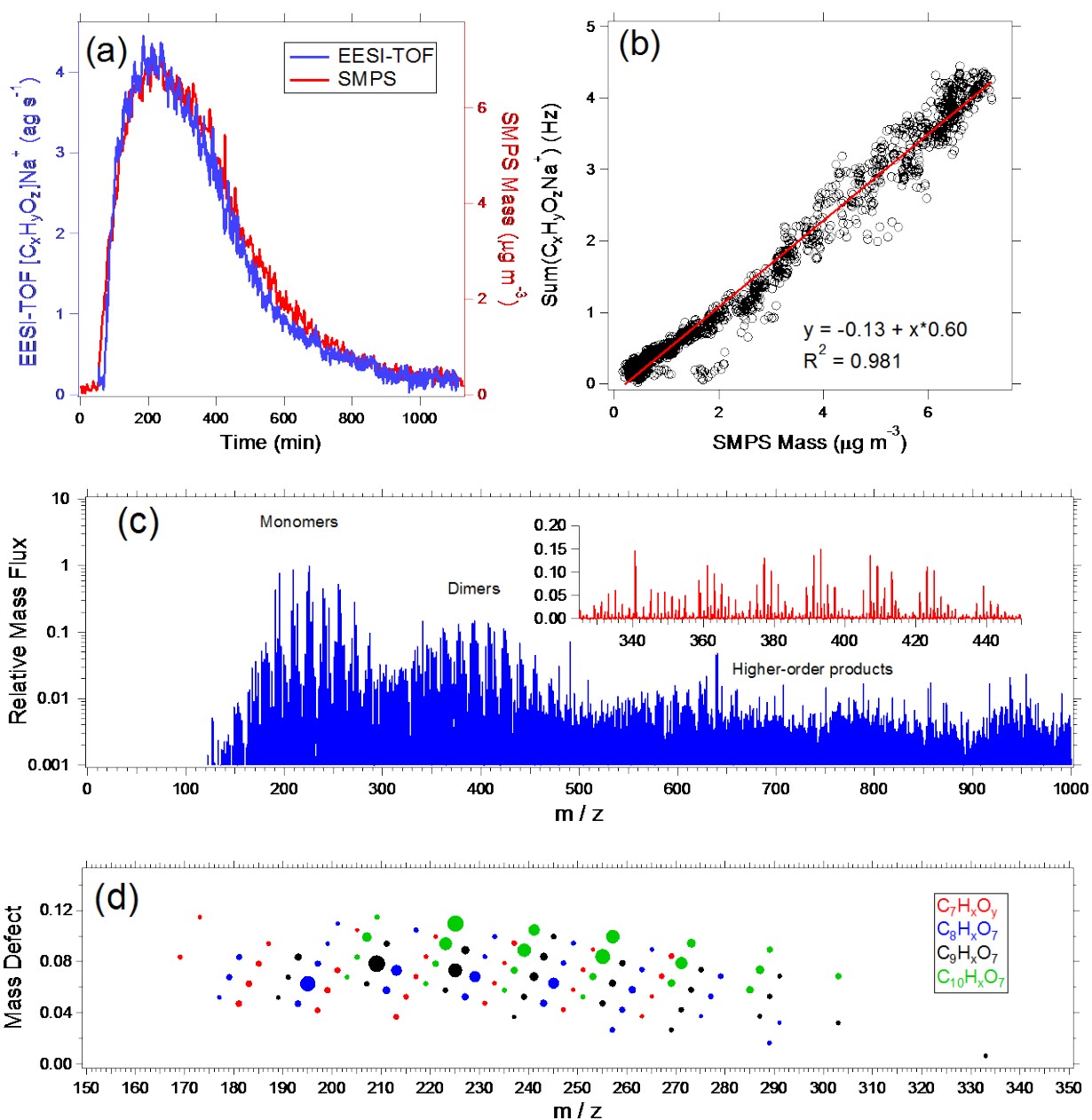

**Figure 9.** Sample EESI-TOF data from α-pinene ozonolysis in an environmental chamber. (a) Summed signal from all $[C_xH_yO_z]Na^+$ ions measured by the EESI-TOF and particle mass measured by the SMPS (effective density = 1.2 g cm$^{-3}$) as a function of time after beginning of reaction. (b) EESI-TOF $[C_xH_yO_z]Na^+$ signal as a function of SMPS mass. (c) Averaged EESI-TOF mass spectrum, with dimer region expanded on linear scale in inset. (d) Mass defect plot of monomer region, with markers sized by signal intensity and colored by number of carbon atoms.

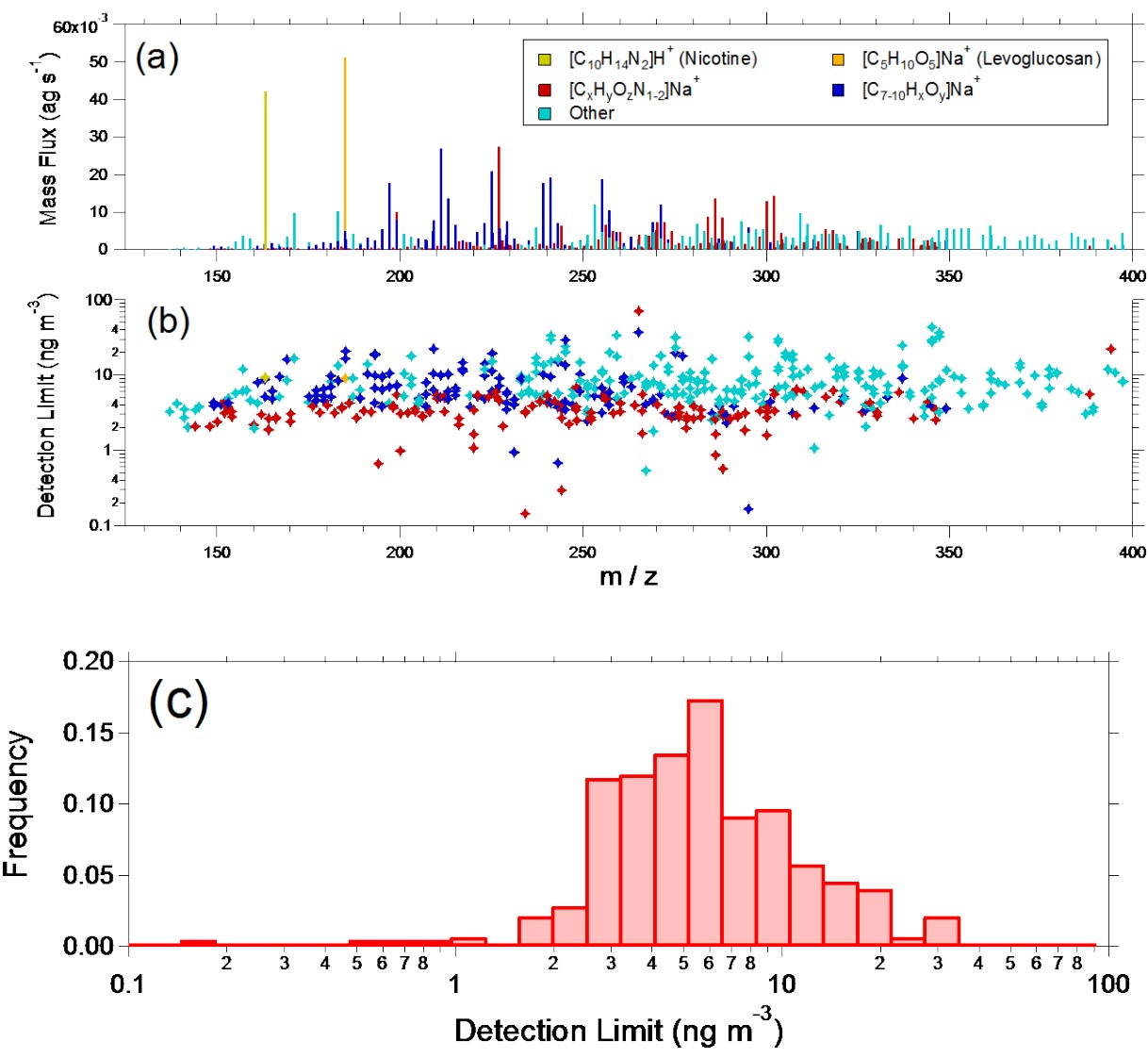

**Figure 10.** (a) Campaign average EESI-TOF mass spectrum from summer measurements in Zurich, Switzerland. Selected ions and families are colored as shown in the legend. Note the x-axis begins at *m/z* 125 due to blanking of smaller ions by the quadrupole ion guides. (b) Campaign median detection limits (30 s), colored as in Fig. 10a. Detection limits are calculated as 3-$\sigma$ variation of the filter blank periods flanking a direct sampling interval, and assuming a uniform sensitivity of 1450 Hz / ($\mu$g m$^{-3}$). (c) Probability distribution of campaign median detection limits.

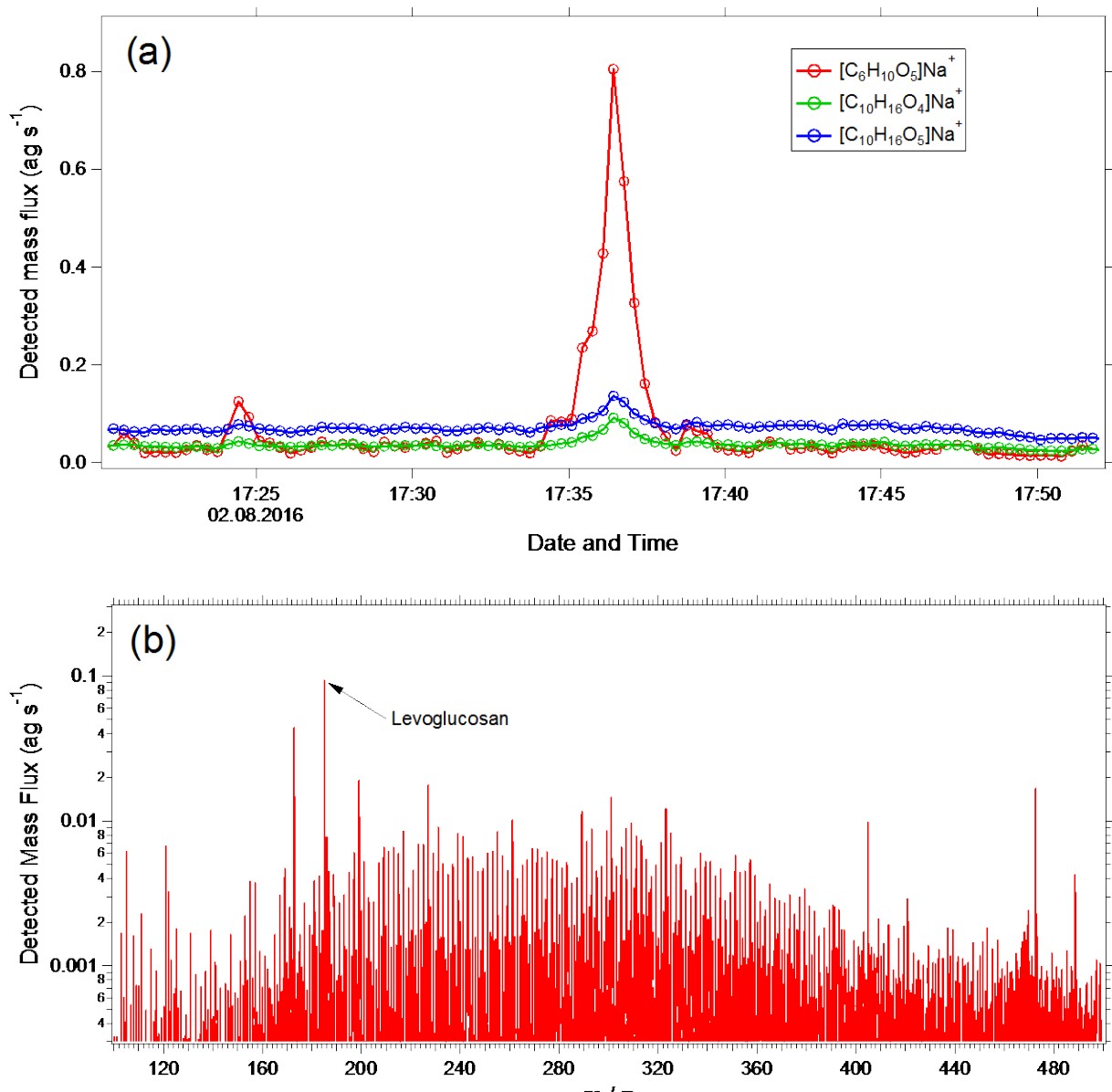

**Figure 11.** (a) Time series of selected ions showing transect of a wildfire plume. Shown are $[C_6H_{10}O_5]Na^+$ (levoglucosan and its isomers) and the monoterpene SOA-influenced ions $[C_{10}H_{16}O_4]Na^+$ and $[C_{10}H_{16}O_5]Na^+$. (b) Mass spectrum (20 s average) during the peak of the wildfire plume with levoglucosan labelled.

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
