# Peer review of "An Extractive Electrospray Ionization Time-of-Flight Mass Spectrometer (EESI-TOF) for online measurement of atmospheric aerosol particles."

_Atmospheric Measurement Techniques, 2019_

## Referee Comment (RC1) · Anonymous Referee #1 · 25 Mar 2019

In their manuscript, Lopez-Hilfiker et al. present the development and characterization of an extractive ESI interface coupled to a TOF-MS. In contrast to previous EESI-MS developments, they demonstrate their setup to have suitable performance, especially regarding detection limits, to allow for real world applications and some proof-of-concept results from deployments at a smog chamber, an aircraft and ambient ground-based measurements are included in the paper as well. There is most certainly a need for better online characterization of particle chemical composition and the described EESI-TOF is a commendable step forward. The paper is generally well written, the

instrument characterization was done in a comprehensive and mostly convincing fashion and the proof-of-concept results provide interesting first insights. I recommend the paper for publication after addressing the following issues.

- The term "near-molecular level" is used a few times. I assume this refers to MS measuring masses only instead of really individual compounds, but the term should be explained in the paper. A word of caution regarding quantitative results from mass measurements vs. fully resolved (chromatography) measurements might be justified as well.

- P7L7-8: What was the flow rate?

- P5L9-10: What does "most species" mean? Which species are not removed and could this pose a problem? Related, could 0.x % denuder breakthrough distort results for volatile species with high gas-phase and very low particle phase concentrations?

- P9-10: I appreciate the discussion on pros and cons of different ESI mixtures, but was a bit surprised to read that MeOH/H2O was used at the end for most studies. My impression from the discussion was that overall ACN/H2O might more suitable, especially due to the stated high background peaks in MeOH/H2O. Please comment on this final choice.

- P13L1-6: Is the usage of mass flux for some and ion flux for most other results really justified? It adds some complexity and the advantage of using mass flux in ag/s is not fully clear to me.

- P14L30 + abstract +conclusions: I find the statement of the "much smaller range" the RRFx spans for SOA as compared to pure components a bit misleading. Given it represents a mean value for a very complex mixture, it is not surprising it varies less than individual compounds. Even more, with Benzene included (its exclusion seems quite arbitrary), the difference to the studied pure model compounds becomes smaller.

- P17L3-4 + abstract +conclusions: The conclusion on the absence of matrix effects

seems to be based on dipentaerythritol experiments only. The insensitivity of this single compound to the specific particle matrix studied cannot, however, be generalized. From conventional ESI it is well known that some species are more susceptible to matrix effects than others. In addition, the organic matrix in these experiments is certainly not representative to the full range of real world particle matrices (both organic and inorganic). I would recommend more caution here. A general absence of any particle matrix effects can only be demonstrated by detailed comparisons with GC/LC-MS based quantification.

- P18L2-5: Can you comment on possible mechanisms of water vapor interference?

---

## Referee Comment (RC2) · Anonymous Referee #3 · 1 May 2019

**Review of Lopez-Hilfiker et al. (2019)**

This manuscript presents results from a newly developed extractive electrospray (EESI) ion source for real-time analysis of organic molecules in aerosol particles. The results are impressive, and the EESI source will be useful for future studies in this area. However, significant clarification is needed before publication, in particular to convince the reader that the performance shown is typical of this source across a wide range of analytes and situations.

Major points

Improved detection limits (DL) is noted throughout the manuscript as a major advance (if not the main one) for this new instrument configuration. However, only a couple general statements are provided regarding DLs (e.g., p13 ln 18-21) without sufficient details to fully understand this key aspect (see additional comment below). I would have expected a subsection in section 3 to be dedicated to showing DLs for a range of elemental formulae or compounds such as those shown in some of the dense example spectra. Also, dependency on sampling time or sampling history, solvent type, compound or m/z and other important aspects that may affect DLs as would be typical in an instrumental characterization paper like this one. Perhaps a mass spectrum of DLs would be appropriate (probably requiring assignment of an approximate fixed sensitivity).

Both mass flux and Hz are used throughout the plots in the paper for signal metrics in various places. Can the authors explain this choice (using one vs the other)? Is it preferred to reflect the mass weighting of the ion signals when showing multiple compounds such as the mass spectra, while not needed for showing single ions? Or if there is not uniform reasoning behind this, perhaps consider using a consistent metric throughout?

Title: the title refers to the EESI-TOF instrument, but the paper is all about the EESI source with no new information or modifications for the TOF mass spectrometer. A TOF does not seem strictly needed either, and EESI has been used before with other mass analyzers. I strongly suggest that the title is updated to reflect this, to something like "An extractive electrospray (EESI) ion source for online mass spectrometric measurements…"

P1 ln 13: DLs without the relevant sampling time are meaningless, please state

P6 ln 3-7: Again, it is unclear what the relevant sampling period for the stated numbers are, please clarify. Strictly speaking, the amount of sample should be stated as well, but given the similarity of the discussed instruments they are probably comparable.

P6 ln 11-14: This statement about "EESI" is unclear whether it refers to previous work or this paper. Probably previous work since it comes before the final paragraph stating what is presented in this paper? If so, provide references?

P7 ln 4-5: It is stated that 1 um Teflon filters are used for blanks. Often for online aerosol instruments, hepa filters with small pressure drops under extended aerosol exposure are used for blanking. Can the authors comment on any issues with pressure differences associated with switching between filter/no-filter during extended sampling with a clean filter and as the filter loads up with aerosol since substantial pressure changes (~20 mbar) might substantially alter the spray. It is noted later (P12 ln 2-3) that only a 2% change in the [NaI]Na+ ion was observed during filter switching during sampling of ~30 ug/m3 of a-pinene ozonolysis SOA, however from that it is not clear how well this type of filtering works under extended operation of polluted air. Did the authors operate with this type of filter for the proof-of-concept tests described in Section 4? It would also seem that using a teflon, low surface area, filter could have less potential artifacts from adsorption/desorption of sticky gases, than HEPA filters with large surface areas composed of glass or plastics. Thus, providing more information/experience on this aspect could be of substantial value for future uses of this source.

P7 ln 7: the gas-phase denuder seems quite small for this application. What is its capacity? Has breakthrough been tested? How often does it need to be replaced, as a function of sampled concentration?

P7 ln 18: "We find that maximum ion transmission is achieved by maximizing the flow rate into the mass spectrometer, which for our pumping configuration is nominally 1 L min-1". It is unclear to me if the authors are simply stating here that sensitivity simply increases with an increasing ion flux into the mass spectrometer, or if there is an additional benefit of the higher flow. I would also be curious to what extent this effect can be separated from the evaporation process changing with flow, and if this was characterized as part of this work.

P 8 ln 6: this statement is too vague. Please provide more information about the estimated temperature or range of temperatures, and the method of estimation.

P 8 ln 19: is this 1 L min-1 STP, or at some reduced pressure going into the MS?

P10 ln 15: how was this binding energy quantified?

P12, ln 24: "In the absence of fragmentation or decomposition, which has not been observed for any system presented herein". It is unclear how the authors reach this conclusion, it is not supported by the data presented. While it is indeed encouraging that the pure compounds measured by EESI-ToF in this manuscript did not decompose during analysis (although none of them are particularly unstable), this certainly cannot be shown for the various types of SOA analyzed, since its compositions is not known otherwise. The similarity of the EESI and FIGAERO results, which are known to be affected by decomposition (Lopez-Hilfiker et al., 2016, cited in the manuscript, also Stark et al., ES&T 2017), suggest that the EESI might also have some degree of decomposition or fragmentation.

More importantly, this study does not present any indirect evidence that the chemical compositions identified by EESI are indeed consistent with no fragmentation. E.g. in the CHARON instrumental manuscript (Mueller et al 2017) a comparison of O:C ratios for bulk (AMS) and CHARON was performed, showing - as expected - a lower O:C in the CHARON consistent with some fragmentation/elimination of oxygenated fragments. Such a comparison for the EESI-ToF for one of the simpler SOA cases would increase confidence that the effects of decomposition and fragmentation are low. In the absence of such supporting evidence, I would strongly qualify the above statement

P13 ln 20-21: Detection limits need associated averaging time information to be meaningful. ~5s, like in Fig. 2?

P13 ln 21-23: Are these comparisons of detection limits of other instruments for similar averaging time?

P 14 ln 29: The authors describe the EESI-ToF sensitivity to benzene SOA as an outlier compared to the other five SOA systems studied. Looking at Fig. 4 the benzene SOA sensitivity is within the variability of EESI-ToF citric acid sensitivity, and the factor of ~20 difference in bulk SOA sensitivity (RRFs 0.05 to 1) is similar to the factor of ~30 spread observed for single compounds. There is also a consistent trend in EESI-ToF sensitivity of SOA produced from the homologous series of benzene, toluene, and trimethylbenzene. Additional clarification is needed to justify considering benzene SOA an outlier and excluding it from the calculation of the variability in EESI-ToF SOA sensitivity.

P15 ln 8-10: Molecular identification would clearly be a major difficulty in doing do this as well. I.e. even if you could just order any compound you wanted, it would be difficult to determine if the isomer detected was the one calibrated for.

P15/L15 - P16/L9: This section on comparison of EESI-ToF vs FIGAERO/I-CIMS raw signal seems a bit underdeveloped and potentially susceptible to misinterpretations. The Hz signals of many compounds are compared between the two instruments which show good correlations overall and among ions series on a log-log basis. It is stated that I-CIMS reaction rates are collision-limited which in conjunction with adduct binding energies dictate sensitivity, which can be operationally estimated by exploring declustering potential (a.k.a. voltage scanning). Based on the good agreement, it is concluded that therefore the EESI-ToF spectra likely reflect the actual distribution of compounds in particles. However, it has been shown that the relative sensitivities for I- can vary widely; therefore there seems to be a large unsubstantiated logical leap here. I worry that readers will interpret the statement that I-CIMS is collision-limited and seemingly glossed over additional factors controlling sensitivity as meaning that the uncalibrated I-CIMS spectrum essentially reflects the relative distributions of compounds in particles.

Why wasn't the voltage scanning sensitivity estimation method used here? This could help close this gap, providing a more direct look at response factor variability for the EESI for a wide range

of atmospheric SOA surrogate compounds? It is especially surprising that this step wasn't taken, given the authors' development of that method. Without calibration or knowing the general instrument sensitivity and declustering settings, it would seem that the slopes don't have much meaning. That said, the good correlation certainly is interesting, useful and promising that the current analysis suggests that similar empirical sensitivities may be derivable/parameterized for the EESI-ToF.

P 15 ln 23: How representative are the 12 m/z's presented in Fig. 5 of the 100+ shown for the same chemical system in Fig. 9? Namely what fraction of the aerosol signal is attributable to these twelve m/z's? Was the FIGAERO comparison extended to other m/z's?

P16, L27-28: How is this known? Analysis of the SMPS distributions? Is the 20 nm seed particle mode separable from the a-pinene ozonolysis nucleation mode? Ensuring that a substantial fraction of the seed is coated seems pretty important to this demonstration. If this is problematic, and nucleation was unavoidable, why weren't larger seed particles used? 20 nm (and growing to 70 nm) seems surprisingly small for this test (and not representative of the size particles in the atmosphere where most of the mass resides).

P17 Section 3.4 (Water vapor dependence): This is a nice analysis. Have the authors considered whether some of the compounds with substantial H2O-dependence effects may be dominated by semi-volatile gases breaking through the denuder? Possibly looking at the humidity-dependence of signals that are enhanced during the filter blanks would help understand that possibility.

P17 ln 19: it would be informative if the authors reported the relative mass fluxes of water from the ESI and from the sample flow (at let's say 50% RH).

P 19 ln 1: Kumbhani et al. report using a 100 μm ID capillary, while this EESI-ToF uses a 50 μm ID capillary. Authors should revise their statement that the Kumbhani et al. capillary has a 5x smaller ID than the EESI-ToF capillary, and revise the subsequent discussion.

P 19 ln 8: please provide a range of expected sizes for the electrospray droplets.

P21 ln 29: Add FIGAERO-CIMS to list of instruments that show thermal decomposition? (see comment above)

P23 ln 7: Change "...for most precursors." to "...for most precursors TESTED."

Figure 5 and discussion: Comparison of several apparent slopes in Fig. 5a with the bars in Fig. 5b seems to not match the relative slopes relationships. E.g. C914H14O7 looks steeper than C914H14O4 and C1016H14O4 but the opposite in Fig. 5b (0.4 vs 0.55 and 0.6). Is this simply due to non-log linear fits being calculated where the regression is highly weighted toward the largest signal data? If that is the case, perhaps also reporting the average ratio (could easily be

added to the same bar plot). Other options may be a log-log fit or other regression fitting methods that don't emphasize the high value data points. The point here is that at least inspection of a log-log plot shows that the relative trends shown in Fig. 5b and discussed in the text may not be robust for each ion but rather reflect just a few of the high-signal points for each ion. On the other hand, if the lower signal points are noisier and closer to DLs, then perhaps it is fine weight those more strongly. Finally, the type of fit (ODR?) and if it was constrained through zero should be reported.

Figure 5 and discussion: It would be useful if the authors reported what fraction of the aerosol signal the sum of all the select ions represents. Most of it? Similarly, it would be useful to show the EESI-ToF and I-CIMS aerosol mass spectra (possibly on a log-scale).

Figure 8b: Please include the particle volume distribution geometric standard deviation on Figure 8b to give a sense of the overlap of sizes, since this experiment was not conducted with monodisperse particles.

Other points

P1, ln 18: Would suggest replacing "SOA compounds"  with "identified SOA components" or similar

P3, ln 23-31: Need references for many of these statements.

P 4 ln 8: My understanding is that instruments like the ATOFMS and PALMS have much more fragmentation of organic molecules than the AMS. The laser ablation instruments often turn organics into C1+, C2+, and ammonium into NO+. However, as worded this section gives the opposite impression.

P5, ln 10-11:  Text states: "Further, there remain fundamental limits to the detection of highly oxidized compounds, as well accretion products for which there is currently no satisfactory online detection technique." It's not clear what is meant by "fundamental limits" which is vague in this context. Please clarify. Also add "as" between "well" and "accretion".

P9, ln 29: This is perhaps a typo? I have not seen water at 25 C to have more than 18.2 MOhms resistance.

P10 ln 28: this sentence is missing a verb

P15 ln 6: a reference to the CIMS strategy described is needed

P16, ln 12-13: Add reference for this statement about matrix effects and ion suppression being common in ESI.

P20 ln 17-18: Remove unneeded "of" and "as" which make the sentence grammatically problematic.

P20 ln12-13 / Fig 9a/b: Please state how mass concentration was determined from SMPS measurements as shown in Fig. 9a/b.

---

## Author Comment (AC1) · 6 Jul 2019

**Response to Reviewer #1**

We thank the reviewer for the helpful comments, which have helped improve the manuscript. Below we provide a point-by-point response to the issues raised by the reviewer. Reviewer comments are provided in *italics*, our responses follow in normal text, and modifications to the manuscript are denoted in blue.

**Comment #1:**

*The term "near-molecular level" is used a few times. I assume this refers to MS measuring masses only instead of really individual compounds, but the term should be explained in the paper. A word of caution regarding quantitative results from mass measurements vs. fully resolved (chromatography) measurements might be justified as well.*

**Response:**

As the reviewer surmises, we use the term "near-molecular level" to indicate that the EESI-TOF can provide a molecular formula but no direct structural information or isomeric separation. We have clarified this term "near-molecular level" in two places:

Abstract: "To address this gap, we present a novel, field-deployable extractive electrospray ionization time-of-flight mass spectrometer (EESI-TOF), which provides online, near-molecular (i.e., molecular formula) OA measurements at atmospherically relevant concentrations without analyte fragmentation or decomposition."

Section 2.1: "The EESI source was developed with the specific goal to measure OA at a near-molecular level (with "near-molecular" here defined as the determination of a molecular formula, without direct structural information or isomeric separation), however it can more generally be applied to gas or combined gas/particle measurements."

We also agree with the reviewer's second point, namely that the possibility of isomers can complicate quantification in experimental or field measurements even if the relevant standards are available. This issue was also raised by Reviewer #2 (Comment #17). We have added the following text to section 3.2, where quantification of EESI-TOF data is considered.

"Further, for aerosol of unknown composition the possibility of multiple isomers (potentially having significantly different $RRF_x$) must be considered."

**Comment #2:**

*P7L7-8: What was the flow rate?*

**Response:**

The flow rate is between 0.7 and 1.0 L min$^{-1}$, depending on the temperature of the inlet capillary. This information has been added to the manuscript.

**Comment #3:**

*What does "most species" mean? Which species are not removed and could this pose a problem? Related, could 0.x % denuder breakthrough distort results for volatile species with high gas-phase and very low particle phase concentrations?*

**Response:**

Questions related to denuder performance were also raised by Reviewer #2 (Comment #8), and our response is duplicated here.

We agree with the reviewer that denuder breakthrough is problematic, as it can result in either increased detection limits or spurious particle signal. These issues were noted in the original manuscript in section 2.1 as motivation for installing the denuder and we leave the original text unchanged except that we now note that these problems can arise not only from the absence of a denuder but also from breakthrough.

We have not conducted a comprehensive, compound-by-compound assessment of the denuder. The term "most species" was instead meant to indicate that for the systems discussed in this manuscript (i.e., pure components, laboratory-generated SOA, and initial ambient measurements in Switzerland), breakthrough has not been observed. More recently, we have experienced some breakthrough issues in heavily polluted ambient locations. Characterization of these effects is ongoing and preliminary results suggest this problem can be solved without degrading performance simply by utilizing a larger denuder.

We now discus this in section 2.1 as follows: "The denuder is very similar to that used for the CHARON-PTRMS, where it removes methanol, acetonitrile, acetaldehyde, acetone, isoprene, methylethylketone, benzene, toluene, xylene, 1,3,5-trimethylbenzene, and α-pinene with >99.9999% efficiency (Eichler et al., 2015). Our experiments show >99.95% removal for pinonic acid, and no detectable breakthrough in the chamber and field experiments presented in section 4, which were conducted at OA concentrations up to approximately 10 µg m$^{-3}$ with the denuder not requiring regeneration for at least 2 weeks. For smog chamber experiments on wood and coal-burning emissions at higher concentrations (20 to 200 µg m$^{-3}$, 1 experiment of 3-4 h per day) (Bertrand et al., submitted), the denuder was regenerated every 2 to 3 days, when a slower response time was observed on switching between the direct sampling and filter blank measurements. This suggests a higher capacity denuder should be used for continuous sampling under polluted conditions."

**Comment #4:**

*P9-10: I appreciate the discussion on pros and cons of different ESI mixtures, but was a bit surprised to read that MeOH/H2O was used at the end for most studies. My impression from the discussion was that overall ACN/H2O might be more suitable, especially due to the stated high background peaks in MeOH/H2O. Please comment on this final choice.*

**Response:**

The main consideration is that operationally the MeOH/H2O spray is easier to stabilize and so was chosen for both the initial characterization experiments presented here and early measurement campaigns (Pospisilova, Stefenelli). With improved machining of the EESI

source and increased user experience, the ACN/H2O mixture has become our preferred working solution for most applications (e.g., Qi) because of its lower background, as correctly identified by the reviewer. As a consequence, the set of characterization experiments performed for the MeOH/H2O system is presently more comprehensive than that of the ACN/H2O system, and we therefore choose to focus on MeOH/H2O results here while noting their comparability to ACN/H2O, rather than the reverse.

**Comment #5:**

*P13L1-6: Is the usage of mass flux for some and ion flux for most other results really justified? It adds some complexity and the advantage of using mass flux in ag/s is not fully clear to me.*

**Response:**

This issue was also raised by Reviewer #2 (Comment #2), and our response is repeated here.

Most studies describe particle-phase composition in terms of mass. Therefore, when discussing total EESI-TOF signal or relative composition, we utilize the mass flux metric. This is the quantity most closely related to mass that we can obtain given the unknown $RRF_x$.

However, a large part of this paper also focuses on the fundamental operation of the EESI-TOF. Here it is desirable to present results in terms of the actual quantity measured (i.e., ion flux), which in our view makes instrument operation/response most transparent. For example, although ionization efficiency and related concepts (Eq. 1 and related discussion), as well as the $RRF_x$, can be in principle discussed in terms of mass, these concepts and the obtained values are most clear when presented in terms of the probabilistic behavior of individual ions and molecules.

We have clarified this in the manuscript as follows (section 3.1):

"Fundamentally, the EESI-TOF measurement is in terms of the ion flux reaching the detector (Hz), as shown on the left axis of Fig. 2. However, in most studies the particle phase is described in terms of mass for both absolute and relative concentrations, making it desirable to also obtain a mass-related metric from the EESI-TOF measurements. In principle, the EESI-TOF ion signal for a molecule $x$ can be converted to a mass concentration according to Eq. (1):"

"For reference, we show on the right axis of Fig. 3 the $MF_x$ (in attograms ($10^{-18}$ g) per second, ag s$^{-1}$) corresponding to the measured $I_x$; however for the remainder of the basic characterization experiments presented herein we show instead the directly measured $I_x$, which is the actual quantity measured by the instrument."

**Comment #6:**

*P14L30 + abstract + conclusions: I find the statement of the "much smaller range" the RRFx spans for SOA as compared to pure components a bit misleading. Given it represents a mean value for a very complex mixture, it is not surprising it varies less than individual compounds.*

*Even more, with Benzene included (its exclusion seems quite arbitrary), the difference to the studied pure model compounds becomes smaller.*

**Response:**

This issue was also raised by Reviewer #1 (Comment #6) and our response is duplicated here.

In the original manuscript, benzene was classified empirically as an outlier because its removal has a considerably larger effect on the range of SOA RRFs (60 % decrease) than does removal of either the highest RRF (1,3,5-trimethylbenzene, 40 % decrease) or the next lowest (phenol, 30 %). However, the low RRF observed for benzene is consistent with the overall observed trend, and we therefore agree it can be misleading to treat it separately. The revised text simply notes that the range of RRFs observed for SOA is lower than that for the pure compounds, while noting that the pure compounds must occupy a larger range than indicated because the RRF for benzene SOA is a factor of 3 lower than any of the pure components. As noted by Reviewer #1, this decreased RRF range for SOA is expected given that it represents the mean RRF of a complex mixture.

The revised text is as follows. "The $RRF_x$ observed for the SOAs span a much smaller range than do the pure components, i.e. a factor 15 between benzene and 1,3,5-trimethylbenzene compared to a factor of ~30 between citric acid and dipentaerythritol (note that the range of $RRF_x$ for pure components is an underestimate, as some pure components must have an $RRF$ at least as low as benzene, which is itself a factor of 3 lower than citric acid). The smaller $RRF_x$ range exhibited by the SOA experiments is expected given that each value represents the mean $RRF$ of a complex mixture and is consistent with ambient observations, where we do not observe major composition-dependent variations in overall EESI-TOF sensitivity to bulk ambient OA (Qi et al., 2019; Stefenelli et al., 2019). However, direct calibration is clearly advisable for compounds for which absolute quantification is desired. Further, for aerosol of unknown composition the possibility of multiple isomers (potentially having significantly different $RRF_x$) must be considered."

The corresponding text in the abstract and conclusions has also been modified.

Abstract: "Although the relative sensitivities to a variety of commercially available organic standards vary by more than a factor of 30, the bulk sensitivity to SOA generated from individual precursor gases varies by only a factor of 15."

Conclusions: "The EESI-TOF sensitivity to SOA generated from a set of individual precursor gases varies within a factor of 15."

**Comment #7:**

*P17L3-4 + abstract + conclusions: The conclusion on the absence of matrix effects seems to be based on dipentaerythritol experiments only. The insensitivity of this single compound to the specific particle matrix studied cannot, however, be generalized. From conventional ESI it is well known that some species are more susceptible to matrix effects than others. In addition, the organic matrix in these experiments is certainly not representative to the full range of real world particle matrices (both organic and inorganic). I would recommend more*

*caution here. A general absence of any particle matrix effects can only be demonstrated by detailed comparisons with GC/LC-MS based quantification.*

**Response:**

We agree that the presence and extent of matrix effects may depend on the identity of the test compound and/or the surrounding matrix. A single case study (dipenaerythritol in SOA α-pinene ozonolysis) is presented in this manuscript; investigation of a large number of systems is beyond the scope of the current work. However, we have also compared the total EESI-TOF OA signal to the detectable fraction of AMS OA (i.e., SOA and oxygenated POA) for ambient aerosol (Qi et al., 2019; Stefenelli et al., 2019), as well as laboratory-generated aerosol that is internally mixed with $(NH_4)_2SO_4$ and/or $NH_4NO_3$. In general, good agreement was observed and discrepancies were explainable by the expected differences in $RRF_x$. Although encouraging, this does not altogether rule out matrix effects for individual OA ions. However, it does indicate that bulk OA sensitivity is likely unaffected by a soluble inorganic matrix.

The following text has been added to section 3.3:

"Although these results are obtained from a single test system, they are consistent with the general trend of matrix effects in EESI studies that are negligible or strongly suppressed relative to conventional electrospray measurements (Chen et al., 2006; Gu et al., 2007; Zhou et al., 2007; Chen et al., 2009). In addition, we find that OA signals measured by the EESI-TOF are well correlated with AMS measurements (e.g. total measurable OA, source apportionment factors, tracer ions) (Qi et al., 2019; Stefenelli et al., 2019). This suggests that EESI-TOF bulk OA measurements are likely not affected by soluble inorganic matrices typical of Central Europe (i.e., internal mixtures with $NH_4NO_3$ and $(NH_4)_2SO_4$ concentrations up to ~10 µg m$^{-3}$), although effects on individual ions cannot be ruled out."

We have also qualified statements in the abstract and conclusions, as follows:

Abstract: "In contrast to conventional electrospray systems, the EESI-TOF response is not significantly affected by a changing OA matrix for the systems investigated."

Conclusions: "We observe an instrument response that is linear with mass and without a detectable dependence on the composition of the OA matrix for a dipentaerythritol/α-pinene SOA test system. Ambient measurements also suggest that bulk OA detection is not significantly affected by a changing matrix of soluble inorganic compounds."

**Comment #8:**

*P18L2-5: Can you comment on possible mechanisms of water vapor interference?*

**Response:**

The following text has been added to section 3.4: "Water can potentially decrease sensitivity by competing with the analyte for $Na^+$ ions (e.g. by displacing the analyte), or increase sensitivity by absorbing energy from $Na^+$-adducts, thereby stabilizing them."

**References**

Bertrand, A., Yuan, B., Stefenelli, G., Qi, L., Pospisilova, V., Tong, Y., Sepideh, E., Huang, R.-J., El Haddad, I., Slowik, J. G., and Prevot, A. S. H.: Characterization of fresh and aged solid fuel combustion organic aerosol by extractive electrospray ionization time-of-flight mass spectrometer (EESI-TOF), Environ. Sci. Technol., submitted.

Chen, H., Venter, A., and Cooks, R. G.: Extractive electrospray ionization for direct analysis of undiluted urine, milk and other complex mixtures without sample preparation, Chem. Commun., 19, 2042-2044, 10.1039/b602614a, 2006.

Chen, H., Gamez, G., and Zenobi, R.: What can we learn from ambient ionization techniques?, J. Am. Soc. Mass Spec., 20, 1947-1963, 2009.

Eichler, P., Müller, M., D'Anna, B., and Wisthaler, A.: A novel inlet system for online chemical analysis of semi-volatile submicron particulate matter, Atmos. Meas. Tech., 8, 1353-1360, 10.5194/amt-8-1353-2015, 2015.

Gu, H. W., Chen, H. W., Pan, Z. Z., Jackson, A. U., Talaty, N., Xi, B. W., Kissinger, C., Duda, C., Mann, D., Raftery, D., and Cooks, R. G.: Monitoring diet effects via biofluids and their implications for metabolomics studies, Anal. Chem., 79, 89-97, 10.1021/ac060946c, 2007.

Qi, L., Chen, M.-D., Stefenelli, G., Pospisilova, V., Tong, Y.-D., Bertrand, A., Hueglin, C., Rigler, M., Ge, X.-L., Baltensperger, U., Prévôt, A. S. H., and Slowik, J. G.: Real-time source quantification of wintertime secondary organic aerosol in Zurich using an extractive electrospray ionization time-of-flight mass spectrometer (EESI-TOF), Atmos. Chem. Phys., 19, 8037-8062, 10.5194/acp-19-8037-2019, 2019.

Stefenelli, G., Lopez-Hilfiker, F. D., Pospisilova, V., Vogel, A., Hüglin, C., Baltensperger, U., Prévôt, A. S. H., and Slowik, J. G.: Source apportionment of ambient organic aerosol by online extractive electrospray ionization time-of-flight mass spectrometry (EESI-TOF), Atmos. Chem. Phys. Discuss., 10.5194/acp-2019-361, 2019.

Zhou, Z. Q., Jin, M., Ding, J. H., Zhou, Y. M., Zheng, J., and Chen, H. W.: Rapid detection of atrazine and its metabolite in raw urine by extractive electrospray ionization mass spectrometry, Metabolomics, 3, 101-104, 10.1007/s11306-006-0050-2, 2007.

---

## Author Comment (AC2) · 6 Jul 2019

**Response to Reviewer #2:**

We thank the reviewer for the thoughtful and detailed review, which has allowed us to improve the manuscript. Below we provide a point-by-point response to the issues raised by the reviewer. Reviewer comments are provided in *italics*, our responses follow in normal text, and modifications to the manuscript are denoted in blue.

**Comment #1:**

*Improved detection limits (DL) is noted throughout the manuscript as a major advance (if not the main one) for this new instrument configuration. However, only a couple general statements are provided regarding DLs (e.g., p13 ln 18-21) without sufficient details to fully understand this key aspect (see additional comment below). I would have expected a subsection in section 3 to be dedicated to showing DLs for a range of elemental formulae or compounds such as those shown in some of the dense example spectra. Also, dependency on sampling time or sampling history, solvent type, compound or m/z and other important aspects that may affect DLs as would be typical in an instrumental characterization paper like this one. Perhaps a mass spectrum of DLs would be appropriate (probably requiring assignment of an approximate fixed sensitivity).*

**Response:**

This is an excellent idea. We now present approximate detection limits determined from the summer field campaign in Zurich in section 4.2. Figure 10 has been modified accordingly. Figure 10a consists of the original Fig. 10 (campaign average mass spectrum, colored by ion type). We have added Fig. 10b, which shows detection limits (30 s) calculated as $3$-$\sigma$ variation of the respective ion signal during the filter blank, and assuming a sensitivity of 1450 cps per $\mu g\ m^{-3}$, corresponding to the estimated bulk sensitivity for SOA during this campaign (Stefenelli et al., 2019). The same color scale is used as in Fig. 10a. Figure 10c shows the results from Fig. 10b in histogram form.

The following discussion addressing the revised Fig. 10 has been added to the manuscript: "Estimated detection limits (30 s) are shown on an ion-by-ion basis in Fig. 10b, using the same color scheme as in Fig. 10a. Detection limits are calculated as the $3$-$\sigma$ variation of the ion signal during the filter blank periods flanking a direct sampling interval, and the campaign median is shown. Detection limits are converted to mass assuming a uniform sensitivity of 1450 Hz / ($\mu g$ $m^{-3}$), which corresponds to the estimated sensitivity of SOA during this campaign (Stefenelli et al., 2019). This is a rough estimate which neglects ion-dependent sensitivities and differences in molecular weight. The results are summarized in histogram form in Fig. 10c. Most species have detection limits in the range 1 to 10 ng $m^{-3}$ (median 5.4 ng $m^{-3}$), with nitrogen-containing ions having slightly lower detection limits than other species. We note that these measurements utilized a methanol:$H_2O$ working solution, and detection limits from the cleaner acetonitrile:$H_2O$ system will likely be lower."

**Comment #2:**

*Both mass flux and Hz are used throughout the plots in the paper for signal metrics in various places. Can the authors explain this choice (using one vs. the other)? Is it preferred to reflect the mass weighting of the ion signals when showing multiple compounds such as the mass spectra, while not needed for showing single ions? Or if there is not uniform reasoning behind this, perhaps consider using a consistent metric throughout?*

**Response:**

This issue was also raised by Reviewer #1 (Comment #5), and our response is repeated here.

Most studies describe particle-phase composition in terms of mass. Therefore, when discussing total EESI-TOF signal or relative composition, we utilize the mass flux metric. This is the quantity most closely related to mass that we can obtain given the unknown $RRF_x$.

However, a large part of this paper also focuses on the fundamental operation of the EESI-TOF. Here it is desirable to present results in terms of the actual quantity measured (i.e., ion flux), which in our view makes instrument operation/response most transparent. For example, although ionization efficiency and related concepts (Eq. 1 and related discussion), as well as the $RRF_x$, can be in principle discussed in terms of mass, these concepts and the obtained values are most clear when presented in terms of the probabilistic behavior of individual ions and molecules.

We have clarified this in the manuscript as follows (section 3.1):

"Fundamentally, the EESI-TOF measurement is in terms of the ion flux reaching the detector (Hz), as shown on the left axis of Fig. 2. However, in most studies the particle phase is described in terms of mass for both absolute and relative concentrations, making it desirable to also obtain a mass-related metric from the EESI-TOF measurements. In principle, the EESI-TOF ion signal for a molecule $x$ can be converted to a mass concentration according to Eq. (1):"

"For reference, we show on the right axis of Fig. 3 the $MF_x$ (in attograms ($10^{-18\text{-}g}$) per second, ag s$^{-1}$) corresponding to the measured $I_x$; however for the remainder of the basic characterization experiments presented herein we show instead the directly measured $I_x$, which is the actual quantity measured by the instrument."

**Comment #3:**

*Title: the title refers to the EESI-TOF instrument, but the paper is all about the EESI source with no new information or modification for the TOF mass spectrometer. A TOF does not seem strictly needed either, and EEIS has been used before with other mass analyzers. I strongly suggest that the title is updated to reflect this, to something like "An extractive electrospray (EESI) ion source for online mass spectrometric measurements…"*

**Response:**

The motivation for the development of this instrument is to fill an important gap in existing OA measurements, namely an online, highly time-resolved, field-deployable system with a

relatively controlled ionization scheme and sufficient detection limits for ambient operation, but without analyte decomposition or fragmentation. We achieved this through the development of a new EESI source, optimized for and coupled to a specific TOF-MS. Although we agree that, in principle, the EESI source presented here can be adapted to other mass spectrometers, it is not clear that all of these capabilities will be preserved. As a simple example, the ion transmission efficiency of the specific TOF-MS used here is an important factor in achieving the requisite detection limits. Likewise, the specific construction of the TOF-MS capillary inlet limits the range of accessible temperatures and time spent at/near atmospheric pressure post-extraction.

As a result, we strongly feel that adaptation of this EESI inlet to a new instrument creates a system which once again requires fundamental and comprehensive characterization, and is thus best considered a separate instrument. To avoid confusion on this point, we therefore prefer to retain the original manuscript title.

We now note this instrument vs. source issue in sensitivity discussion in Section 3.2: "Note that these improved detection limits represent the performance of the entire EESI-TOF instrument relative to previous instruments, and we cannot disentangle the contributions of the ionization unit and MS detector."

**Comment #4:**

*P1 ln 13: DLs without the relevant sampling time are meaningless, please state.*

**Response:**

The revised text reads: "The EESI-TOF achieves a linear response to mass, with detection limits on the order of 1 to 10 ng m$^{-3}$ in 5 s for typical atmospherically-relevant compounds."

**Comment #5:**

*P6 ln 3-7: Again, it is unclear what the relevant sampling period for the stated numbers are, please clarify. Strictly speaking, the amount of sample should be stated as well, but given the similarity of the discussed instruments they are probably comparable.*

**Response:**

We agree that this information would be useful, however it is not available from the cited literature. We are able to provide integration times for the Gallimore et al. system, however: "A more recent EESI system (Gallimore and Kalberer, 2013; Gallimore et al., 2017) attained individual compound detection limits of ~1 µg m$^{-3}$ for 100 s integration (M. Kalberer, private communication), which were further reduced to as low as 0.25 μg m$^{-3}$ using tandem mass spectrometry (MS$^2$) and collisionally-induced dissociation (CID)."

**Comment #6:**

*P6 ln 11-14: This statement about "EESI" is unclear whether it refers to previous work or this paper. Probably previous work since it comes before the final paragraph stating what is presented in this paper? If so, provide references?*

**Response:**

This refers to the Gallimore and Kalberer (2013) study discussed earlier in the paragraph. The reference is now repeated for clarity.

**Comment #7:**

*P7 ln 4-5: It is stated that 1 um Teflon filters are used for blanks. Often for online aerosol instruments, hepa filters with small pressure drops under extended aerosol exposure are used for blanking. Can the authors comment on any issues with pressure differences associated with switching between filter/no-filter during extended sampling with a clean filter and as the filter loads up with aerosol since substantial pressure changes (~20 mbar) might substantially alter the spray. It is noted later (P12 ln 2-3) that only a 2% change in the [NaI]Na$^+$ ion was observed during filter switching during sampling of ~30 ug/m3 of a-pinene ozonolysis SOA, however from that it is not clear how well this type of filtering works under extended operation of polluted air. Did the authors operate with this type of filter for the proof-of-concept tests described in Section 4? It would also seem that using a Teflon, low surface area, filter could have less potential artifacts from adsorption/desorption of sticky gases, than HEPA filters with large surface areas composed of glass or plastics. Thus, providing more information/experience on this aspect could be of substantial value for future users of this source.*

**Response:**

The 1 µm Teflon filters were used for the laboratory characterization experiments. The measurements described in section 4 instead used a nylon cartridge filter (Parker Balston, 9933-11-BQ), which is now noted in the manuscript. These two options provided very similar performance. A HEPA filter was tested in one very recent campaign, but evaluation of its performance is ongoing. Regardless of filter type, we expect that some amount of absorption/desorption is inevitable, however these artifacts are removed by a properly functioning denuder.

The magnitude of the change in the [NaI]Na$^+$ signal depends primarily on the details of spray optimization. Thereafter, during stable operation (i.e. without clogging or user adjustments) we have not observed this value to change in response to increased filter loading. However, during high concentration studies the filter is changed regularly to minimize challenge to the denuder by desorbing semivolatile material, and this maintenance schedule may also prevent us from observing the suggested effect.

The following text has been added to the manuscript: "For the proof-of-concept deployments discussed in Section 4, this filter was replaced with a nylon cartridge filter (9933-11-NQ, Parker Balston, Lancaster, NY, USA), which performed similarly."

**Comment #8:**

*P7 ln 7: the gas-phase denuder seems quite small for this application. What is its capacity? Has breakthrough been tested? How often does it need to be replaced, as a function of sampled concentration?*

**Response:**

Questions related to denuder performance were also raised by Reviewer #1 (Comment #3), and our response is duplicated here.

We agree with the reviewer that denuder breakthrough is problematic, as it can result in either increased detection limits or spurious particle signal. These issues were noted in the original manuscript in section 2.1 as motivation for installing the denuder and we leave the original text unchanged except that we now note that these problems can arise not only from the absence of a denuder but also from breakthrough.

We have not conducted a comprehensive, compound-by-compound assessment of the denuder. The term "most species" was instead meant to indicate that for the systems discussed in this manuscript (i.e., pure components, laboratory-generated SOA, and initial ambient measurements in Switzerland), breakthrough has not been observed. More recently, we have experienced some breakthrough issues in heavily polluted ambient locations. Characterization of these effects is ongoing and preliminary results suggest this problem can be solved without degrading performance simply by utilizing a larger denuder.

We now discus this in section 2.1 as follows: "The denuder is very similar to that used for the CHARON-PTRMS, where it removes methanol, acetonitrile, acetaldehyde, acetone, isoprene, methylethylketone, benzene, toluene, xylene, 1,3,5-trimethylbenzene, and α-pinene with >99.9999% efficiency (Eichler et al., 2015). Our experiments show >99.95% removal for pinonic acid, and no detectable breakthrough in the chamber and field experiments presented in section 4, which were conducted at OA concentrations up to approximately 10 µg m$^{-3}$ with the denuder not requiring regeneration for at least 2 weeks. For smog chamber experiments on wood and coal-burning emissions at higher concentrations (20 to 200 µg m$^{-3}$, 1 experiment of 3-4 h per day) (Bertrand et al., submitted), the denuder was regenerated every 2 to 3 days, when a slower response time was observed on switching between the direct sampling and filter blank measurements. This suggests a higher capacity denuder should be used for continuous sampling under polluted conditions."

**Comment #9:**

*P7 ln 18: "We find that maximum ion transmission is achieved by maximizing the flow rate into the mass spectrometer, which for our pumping configuration is nominally 1 L min-1". It is unclear to me if the authors are simply stating here that sensitivity simply increases with an increasing ion flux to the mass spectrometer, or if there is an additional benefit of the higher flow. I would also be curious to what extent this effect can be separated from the evaporation process changing with flow, and if this was characterized as part of this work.*

**Response:**

We have empirically observed that the increased ion transmission at high flow rate is greater than can be explained solely by an increase proportional to flow rate. However, this is at present purely an empirical observation, and we are unable to provide further insight into the mechanism.

**Comment #10:**

*P8 ln 6: This statement is too vague. Please provide more information about the estimated temperature or range of temperatures, and the method of estimation.*

**Response:**

Unfortunately, we are unable to be more precise here because we lack the capability to either measure the gas temperature at the capillary outlet or to model heat transfer throughout the entire inlet system. Although admittedly less informative than we would like, we do consider the current statement accurate given the short residence time and the fact that the 240 °C temperature is measured at the heating unit rather than the actual capillary walls. This latter point was not in the original manuscript, but has been added.

**Comment #11:**

*P8 ln 19: is this 1 L min-1 STP, or at some reduced pressure going into the MS?*

**Response:**

We have clarified that this flow rate is at STP.

**Comment #12:**

*P10 ln 15: how was this binding energy quantified?*

**Response:**

We apologize for omitting the references (Rodgers and Armentrout, 1999; Pejov, 2002), which in turn rely on a combination of quantum chemical calculations and collision-induced dissociation experiments.

**Comment #13:**

*P 12, ln 24: "In the absence of fragmentation or decomposition, which has not been observed for any system presented herein". It is unclear how the authors reach this conclusion, it is not supported by the data presented. While it is indeed encouraging that the pure compounds measured by the EESI-TOF in this manuscript did not decompose during analysis (although none of them are particularly unstable), this certainly cannot be shown for the various types of SOA analyzed, since its composition is not known otherwise. The similarity of the EESI and FIGAERO results, which are known to be affected by decomposition (Lopez-Hilfiker et al.,*

*2016, cited in the manuscript, also Stark et al., ES&T 2017), suggest that the EESI might also have some degree of decomposition or fragmentation.*

*More importantly, this study does not present any indirect evidence that the chemical compositions identified by EESI are indeed consistent with no fragmentation. E.g. in the CHARON instrumental manuscript (Mueller et al., 2017) a comparison of O:C ratios for bulk (AMS) and CHARON was performed, showing – as expected – a lower O:C in the CHARON consistent with some fragmentation/elimination of oxygenated fragments. Such a comparison for the EESI-TOF for one of the simpler SOA cases would increase confidence that the effects of decomposition and fragmentation are low. In the absence of such supporting evidence, I would strongly qualify the above statement.*

**Response:**

We agree that the text highlighted by the reviewer gives the impression that we have performed a detailed assessment of thermal decomposition for every aerosol measured in the paper, which overstates the extent of characterization. The revised text, which is concerned primarily with quantification of individual ions, simply begins: "Assuming no fragmentation or decomposition".

In response to the larger point raised by the reviewer, we provide here some additional information relevant to the understanding of possible thermal decomposition in the EESI-TOF. As noted by the reviewer, we can verify that the pure compounds tested herein did not decompose. This rules out decomposition of aerosol less thermally stable than citric acid (which decomposes at approximately 200 °C).

We agree with the reviewer regarding the difficulty associated with verifying a lack of thermal decomposition in complex aerosol. One way to address this is to ramp the temperature of the capillary. However, as decreasing the capillary temperature increases the instrument gas load, an operational limit is reached when the temperature of the cartridge heaters remain well above room temperature. Our ability to test hotter temperatures is also limited by hardware constraints on heater operation. Nonetheless, over the range of accessible temperatures, we have observed an increase in sensitivity to α-pinene SOA across the entire mass spectrum with increasing temperature. In the event of thermal decomposition, a decrease in apparent sensitivity would be expected, whereas the increased sensitivity is likely due to more droplet evaporation. Although we cannot rule out the possibility that thermal decomposition occurs at levels small relative to the sensitivity increase, these results do suggest that it is at most a minor effect.

The reviewer also raises comparisons of the EESI-TOF and FIGAERO operation, as potential comparisons between EESI-TOF and AMS atomic ratios. Regarding the EESI-TOF vs. FIGAERO, we note that although the unit temperatures in question are relatively similar, exposure to these temperatures in the EESI-TOF is on the order of $10^5$-$10^6$ times shorter in the EESI-TOF, greatly reducing heat transfer.

Comparison of the atomic O:C ratios between EESI-TOF and AMS have been performed for summer and winter campaigns in Zurich (Qi et al., 2019; Stefenelli et al., 2019). In all cases the EESI-TOF values are greater than or equal to the AMS. However, while ion-molecule reaction rates and thus compound-dependent response factors are well-constrained in the CHARON-PTR system, compound-dependent EESI-TOF extraction efficiencies and $RRF_x$

have not been characterized in detail and may strongly affect the comparison. Therefore we do not believe this data can be used to check for thermal decomposition in the EESI-TOF.

**Comment #14:**

*P13 ln 20-21: Detection limits need associated averaging time information to be meaningful. ~5s, like in Fig. 2?*

**Response:**

Yes, this refers to 5 s measurements (added to the manuscript). Note, however, that we state that 10 ng m$^{-3}$ are "readily detectable" with 5 s of averaging, not that the 5 s detection limits are 10 ng m$^{-3}$ (they are lower).

**Comment #15:**

*P 13 ln 21-23: Are these comparisons of detection limits for other instruments for similar averaging time?*

**Response:**

See response to Comment #5. Averaging times are not reported in the cited manuscripts, although we are able to provide them for the Gallimore et al. system. However, based on the sample data provided in these studies, the times for the other instruments are likely equal or longer.

**Comment #16:**

*P 14 ln 29: The authors describe the EESI-TOF sensitivity to benzene SOA as an outlier compared to the other five SOA systems studied. Looking at Fig. 4 the benzene SOA sensitivity is within the variability of EESI-TOF citric acid sensitivity, and the factor of ~20 difference in bulk SOA sensitivity (RRFs 0.05 to 1) is similar to the factor of ~30 spread observed for single compounds. There is also a consistent trend in EESI-TOF sensitivity of SOA produced from the homologous series of benzene, toluene, and trimethylbenzene. Additional clarification is needed to justify considering benzene SOA an outlier and excluding it from the calculation of the variability in EESI-TOF SOA sensitivity.*

**Response:**

This issue was also raised by Reviewer #1 (Comment #6) and our reponse is duplicated here.

In the original manuscript, benzene was classified empirically as an outlier because its removal has a considerably larger effect on the range of SOA RRFs (60 % decrease) than does removal of either the highest RRF (1,3,5-trimethylbenzene, 40 % decrease) or the next lowest (phenol, 30 %). However, the low RRF observed for benzene is consistent with the overall observed trend, and we therefore agree it can be misleading to treat it separately. The revised text simply notes that the range of RRFs observed for SOA is lower than that for the pure compounds, while noting that the pure compounds must occupy a larger range than indicated because the

RRF for benzene SOA is a factor of 3 lower than any of the pure components. As noted by Reviewer #1, this decreased RRF range for SOA is expected given that it represents the mean RRF of a complex mixture.

The revised text is as follows. "The $RRF_x$ observed for the SOAs span a much smaller range than do the pure components, i.e. a factor 15 between benzene and 1,3,5-trimethylbenzene compared to a factor of ~30 between citric acid and dipentaerythritol (note that the range of $RRF_x$ for pure components is an underestimate, as some pure components must have an $RRF$ at least as low as benzene, which is itself a factor of 3 lower than citric acid). The smaller $RRF_x$ range exhibited by the SOA experiments is expected given that each value represents the mean $RRF$ of a complex mixture and is consistent with ambient observations, where we do not observe major composition-dependent variations in overall EESI-TOF sensitivity to bulk ambient OA (Qi et al., 2019; Stefenelli et al., 2019). However, direct calibration is clearly advisable for compounds for which absolute quantification is desired. Further, for aerosol of unknown composition the possibility of multiple isomers (potentially having significantly different $RRF_x$) must be considered."

The corresponding text in the abstract and conclusions has also been modified.

Abstract: "Although the relative sensitivities to a variety of commercially available organic standards vary by more than a factor of 30, the bulk sensitivity to SOA generated from individual precursor gases varies by only a factor of 15."

Conclusions: "The EESI-TOF sensitivity to SOA generated from a set of individual precursor gases varies within a factor of 15."

**Comment #17:**

*P15 ln 8-10: Molecular identification would clearly be a major difficulty in doing this as well. I.e., even if you could just order any compound you wanted, it would be difficult to determine if the isomer detected was the one calibrated for.*

**Response:**

We agree, and have added the following text:

"Further, for aerosol of unknown composition the possibility of multiple isomers (potentially having significantly different $RRF_x$) must be considered."

**Comment #18:**

*P15/L15 – P16/L9: This section on comparison of EESI-TOF vs. FIGAERO/I-CIMS raw signal seems a bit underdeveloped and potentially susceptible to misinterpretations. The Hz signals of many compounds are compared between the two instruments which show good correlations overall and among ions on a log-log basis. It is stated that I-CIMS reaction rates are collision-limited which in conjunction with adduct binding energies dictate sensitivity, which can be operationally estimated by exploring declustering potential (a.k.a. voltage scanning). Based*

*on the good agreement, it is concluded that therefore the EESI-TOF spectra likely reflect the actual distribution of compounds in particles. However, it has been shown that the relative sensitivities for I- can vary widely; therefore there seems to be a large unsubstantiated logical leap here. I worry that readers will interpret the statement that I-CIMS is collision-limited and seemingly glossed over additional factors controlling sensitivity as meaning that the uncalibrated I-CIMS spectrum essentially reflects the relative distributions of compounds in particles.*

*Why wasn't the voltage scanning sensitivity estimation method used here? This could help close this gap, providing a more direct look at response factor variability for the EESI for a wide range of atmospheric SOA surrogate compounds? It is especially surprising that this step wasn't taken, given the authors' development of that method. Without calibration or knowing the general instrument sensitivity and declustering settings, it would seem that the slopes don't have much meaning. That said, the good correlation certainly is interesting, useful and promising that the current analysis suggests that empirical sensitivities may be derivable/parameterized for the EESI-TOF.*

**Response:**

We agree with the reviewer that a detailed comparison of the EESI-TOF with the FIGAERO-I-CIMS (or other related instruments) is of high interest, and regarding the potential utility of the voltage scanning for sensitivity estimation. However, these investigations are very complex, whereas the measurements performed for this comparison were intended only as a basic proof-of-principle test with the specific goal of demonstrating that the EESI-TOF response is qualitatively similar to a gas-phase CIMS despite the additional complications of condensed-phase extraction and droplet evaporation/ionization. Because of this limited goal, we did not conduct the voltage scanning suggested by the reviewer and a relatively limited amount of data is available. We have since conducted several dedicated campaigns focused on the inter-comparison of EESI-TOF with related instruments, and prefer to defer a detailed discussion of such inter-comparisons to the presentation of these studies.

The reviewer also identifies the final sentence in section 3.2 ("Further, the correlation [between FIGAERO and EESI-TOF] indicates that the spectral patterns observed in the EESI-TOF likely reflect to a large degree the actual distribution of compounds in the particle phase" as prone to misinterpretation. We agree, and have removed this sentence. Note also that a detailed discussion of the factors governing I-CIMS sensitivity was presented in the preceding paragraph in the original manuscript.

**Comment #19:**

*P15 ln 23: How representative are the 12 m/z's presented in Fig. 5 of the 100+ shown for the same chemical system in Fig. 9? Namely what fraction of the aerosol signal is attributable to these twelve m/z's? Was the FIGAERO comparison extended to other m/z's?*

**Response:**

The ion series selected for comparison ($C_9H_{14}O_x$ and $C_{10}H_{16}O_x$) together comprise 21% of the EESI-TOF signal and 19% of the particle-phase FIGAERO-I-CIMS signal. This information has been added to section 3.2. As noted in the previous response, the FIGAERO/EESI-TOF

comparison was intended only as a basic proof-of-principle test, and we prefer to defer detailed comparisons of the mass spectra to the dedicated inter-comparison campaigns that have since been conducted.

**Comment #20:**

*P16, L27-28: How is this known? Analysis of SMPS distributions? Is the 20 nm seed particle mode separable from the a-pinene ozonolysis nucleation mode? Ensuring that a substantial fraction of the seed is coated seems pretty important to this demonstration. If this is problematic, and nucleation was unavoidable, why weren't larger seed particles used? 20 nm (and growing to 70 nm) seems surprisingly small for this test (and not representative of the size of particles in the atmosphere where most of the mass resides).*

**Response:**

The fraction of coated particles was determined from analysis of the SMPS distributions; this information has been added to the manuscript.

Regarding the selected seed particle size, this reflects the mode of the number distribution generated from our nebulizer. While larger particles would be indeed be desirable to better represent the atmospheric mass mode, we have observed that size-selecting particles with a DMA greatly reduces their detection efficiency by the EESI-TOF (even after passing through a second nebulizer). The reasons for this are not yet clear, but prevented us from attempting the suggested experiments.

We additionally note that good agreement between the total EESI-TOF signal and the relevant fraction of AMS OA (SOA and oxygenated POA) have been obtained for a variety of systems, including ambient accumulation mode particles. This information was added to section 3.3 as follows: "In addition, we find that OA signals measured by the EESI-TOF are well correlated with AMS measurements (e.g. total measurable OA, source apportionment factors, tracer ions) (Qi et al., 2019; Stefenelli et al., 2019). This suggests that EESI-TOF bulk OA measurements are likely not affected by soluble inorganic matrices typical of Central Europe (i.e., internal mixtures with $NH_4NO_3$ and $(NH_4)_2SO_4$ concentrations up to ~10 µg m$^{-3}$), although effects on individual ions cannot be ruled out."

**Comment #21:**

*P17 Section 3.4 (Water vapor dependence): This is a nice analysis. Have the authors considered whether some of the compounds with substantial H2O-dependence effects may be dominated by semi-volatile gases breaking through the denuder? Possibly looking at the humidity-dependence of signals that are enhanced during the filter blanks would help understand that possibility.*

**Response:**

In these experiments, we have not identified any signal attributable to denuder breakthrough (aside from water clusters with ions derived from the working solution or NaI dopant, which are not relevant to the question posed).

**Comment #22:**

*P17 ln 19: it would be informative if the authors reported the relative mass fluxes of water from the ESI and from the sample flow (at let's say 50% RH).*

**Response:**

We assume a sample flow rate of 0.8 L min$^{-1}$ at 25 °C, corresponding to a molecular flux of 2.27 x 10$^{20}$ molec min$^{-1}$ of H$_2$O. We do not directly measure the flowrate through the ESI capillary, but this is estimated to be in the range of 1 to10 µL min$^{-1}$. Given a 1:1 mixture of water and either acetonitrile or methanol as a working solution, we estimate the H$_2$O flux to be 1.67 x 10$^{19}$ to 1.67 x 10$^{20}$ molec min$^{-1}$ through the electrospray capillary. Therefore the ESI capillary provides approximately 7 to 42 % of the water at 50% RH.

The following text has been added to section 3.4: "Assuming a sample flow of 0.8 L min$^{-1}$ and 1 to 10 µL min$^{-1}$ flow through the electrospray capillary, the working solution provides 7 to 42 % of the total water at 50 % RH."

**Comment #23:**

*P19 ln 1: Kumbhani et al. report using a 100 µm ID capillary, while this EESI-TOF uses a 50 µm ID capillary. Authors should revise their statement that the Kumbani et al. capillary has a 5x smaller ID than the EESI-TOF capillary, and revise the subsequent discussion.*

**Response:**

We apologize for the error. The incorrect reference to the dimensions of the Kumbhani et al. capillary has been removed. The remainder of the discussion merely speculates on reasons for the different behavior observed in the two systems, which requires follow-up investigations to resolve and is therefore left unchanged.

**Comment #24:**

*P19 ln 8: please provide a range of expected sizes for the electrospray droplets*

**Response:**

We do not have the capability at present to directly measure the electrospray droplet size distribution. Based on a literature comparison of roughly similar sources, we estimate a droplet mode on the order of 4 to 40 µm. This information has been added to the manuscript.

**Comment #25:**

*P21 ln 29: Add FIGAERO-CIMS to the list of instruments that show thermal decomposition? (see comment above).*

**Response:**

The revised text reads: "As discussed above, current OA measurements used in mobile measurements require a tradeoff between thermal decomposition (extensive for the AMS, minor for FIGAERO-CIMS), ionization-induced fragmentation (AMS, CHARON-PTRMS), or time resolution (FIGAERO-CIMS)."

**Comment #26:**

*P23 ln 7: Change "…for most precursors." To "…for most precursors TESTED."*

**Response:**

This sentence was revised in response to Comment #16 and Reviewer #1 (Comment #6) and now reads: "The EESI-TOF sensitivity to SOA generated from a set of individual precursor gases varies within a factor of 15."

**Comment #27:**

*Figure 5 and discussion: Comparison of several apparent slopes in Fig. 5a with the bars in Fig. 5b seems not to match the relative slopes relationships. E.g. $C9H14O7$ looks steeper than $C9H14O4$ and $C10H14O4$ but the opposite in Fig. 5b (0.4 vs. 0.55 and 0.6). Is this simply due to non-log linear fits being calculated where the regression is highly weighted toward the largest signal data? If that is the case, perhaps also reporting the average ratio (could easily be added to the same bar plot). Other options may be a log-log fit or other regression fitting methods that don't emphasize the high value data points. The point here is that at least inspection of a log-log plot shows that the relative trends shown in Fig. 5b and discussed in the text may not be robust for each ion but rather reflect just a few of the high-signal points for each ion. On the other hand, if the lower signal points are noisier and closer to DLs, then perhaps it is fine to weight those more strongly. Finally, the type of fit (ODR?) and if it was constrained through zero should be reported.*

**Response:**

The apparent discrepancy between the visual appearance of slopes and their actual value is due to non-zero intercepts that are somewhat different for each ion. Lines having the same slope but different intercepts diverge at low values on a log-log plot, with a larger x-intercept corresponding to a visually steeper (but numerically identical) slope. The fits are not significantly influenced by outliers, and must be fit in linear space because they depend on absolute ion counts and not ratio of ion counts to background (as would be the case for a log-log fit). Data are plotted on a logarithmic scale only because ion signals span several orders of magnitude. As a result, we have not altered Fig. 5. However, we now state in the figure caption that all fits were conducted using ODR and that the intercept is unconstrained.

**Comment #28:**

*Figure 5 and discussion: It would be useful if the authors reported what fraction of the aerosol signal the sum of all the selected ions represents. Most of it? Similarly, it would be useful to show the EESI-TOF and I-CIMS aerosol mass spectra (possibly on a log-scale).*

**Response:**

The requested figure has been added as Fig. 5a. We compare the range $m/z$ 100 to 500 for both instruments, with $Na^+$ and $I^-$ subtracted from the EESI-TOF and FIGAERO-I-CIMS, respectively. The ions selected for comparison ($C_9H_{14}O_x$ and $C_{10}H_{16}O_x$ families) comprise 21 % of the EESI-TOF signal and 19 % of the FIGAERO signal.

The revised text follows: "Figure 5 shows the EESI-TOF and FIGAERO-I-CIMS mass spectra from $m/z$ 100 to 500 after subtraction of $Na^+$ and $I^-$, respectively. Spectra are normalized such that the sum across the selected $m/z$ range is 1. The main features and general trends are similar between the two instruments, although some differences in relative intensity are evident. In addition, the EESI-TOF signal is greatly reduced below $m/z$ 150 due to the selected transmission window in the quadrupole ion guides. Figure 5b shows EESI-TOF signals as a function of the FIGAERO-I-CIMS for selected ions, specifically the $[C_9H_{14}O_x]Na^+$ (red) and $[C_{10}H_{16}O_x]Na^+$ (blue) series, which constitute 21% (EESI-TOF) and 19% (FIGAERO-I-CIMS) of the total particle-phase signal."

**Comment #29:**

*Figure 8b: Please include the particle volume distribution standard deviation on Fig. 8b to give a sense of the overlap of sizes, since this experiment was not conducted with monodisperse particles.*

**Response:**

We have added error bars to Fig. 8b, where the error bars denote the mobility diameters corresponding to the half-maxima of the polydisperse particle volume distribution.

**Comment #30:**

*P1, ln 18: Would suggest replacing "SOA compounds" with "identified SOA components" or similar.*

**Response:**

The revised text reads: "…the bulk sensitivity to SOA generated from most tested precursors varies by only a factor of 6."

**Comment #31:**

*P3, ln 23-31: Need references for many of these statements.*

**Response:**

References have been added.

**Comment #32:**

*P4 ln 8: My understanding is that instruments like the ATOFMS and PALMS have much more fragmentation of organic molecules than the AMS. The laser ablation instruments often turn organics into C1+, C2+, and ammonium into NO+. However, as worded this section gives the opposite impression.*

**Response:**

ATOFMS has been used successfully to measure oligomers (Gross et al., 2006), which decompose and/or fragment in the AMS. Therefore at least some combination of compounds/instruments can provide reduced fragmentation/decomposition relative to the AMS, and the text is left unchanged.

**Comment #33:**

*P5 ln 10-11: Text states: "Further, there remain fundamental limits to the detection of highly oxidized compounds, as well accretion products for which there is currently no satisfactory online detection technique." It's not clear what is meant by "fundamental limits" which is vague in this context. Please clarify. Also add "as" between "well" and "accretion".*

**Response:**

We agree that "fundamental" was unclear, and it has been deleted. The identified typo is fixed.

**Comment #34:**

*P9 ln 29: This is perhaps a typo? I have not seen water at 25 C to have more than 18.2 MOhms resistance.*

**Response:**

The typo has been corrected (should have been 18.2 M$\Omega$ cm).

**Comment #35:**

*P10 ln 28: this sentence is missing a verb.*

**Response:**

The corrected sentence reads: "Preliminary investigations using an $H_2O$-only working fluid (with NaI dopant) were also conducted."

**Comment #36:**

*P15 ln 6: a reference to the CIMS strategy described is needed.*

**Response:**

This sentence concludes the discussion of EESI-TOF sensitivity to bulk laboratory-generated SOA and pure compounds, for which the SMPS is used as a reference. Discussion of the CIMS does not begin until the following paragraph. We therefore prefer to leave the text unchanged.

**Comment #37:**

*P16, ln 12-13: Add reference for this statement about matrix effects and ion suppression being common in ESI.*

**Response:**

References added.

**Comment #38:**

*P20 ln 17-18: Remove unneeded "of" and "as" which make the sentence grammatically problematic.*

**Response:**

The revised text reads: "To aid the eye, $[(NaI)_n]Na^+$ clusters are removed; this is done because although the background-subtracted $[(NaI)_n]Na^+$ is close to zero, it is the difference of two high-intensity signals and therefore remains large relative to most ions in the mass spectrum."

**Comment #39:**

*P20 ln12-13 / Fig 9a/b: Please state how mass concentration was determined from SMPS measurements as shown in Fig. 9a/b.*

**Response:**

We assumed an effective density of 1.2 g cm$^{-3}$.

**References**

Bertrand, A., Yuan, B., Stefenelli, G., Qi, L., Pospisilova, V., Tong, Y., Sepideh, E., Huang, R.-J., El Haddad, I., Slowik, J. G., and Prevot, A. S. H.: Characterization of fresh and aged solid fuel combustion organic aerosol by extractive electrospray ionization time-of-flight mass spectrometer (EESI-TOF), Environ. Sci. Technol., submitted.

Eichler, P., Müller, M., D'Anna, B., and Wisthaler, A.: A novel inlet system for online chemical analysis of semi-volatile submicron particulate matter, Atmos. Meas. Tech., 8, 1353-1360, 10.5194/amt-8-1353-2015, 2015.

Gallimore, P. J., and Kalberer, M.: Characterizing an extractive electrospray ionization (EESI) source for the online mass spectrometry analysis of organic aerosols, Environ. Sci. Technol., 47, 7324-7331, 10.1021/es305199h, 2013.

Gallimore, P. J., Giorio, C., Mahon, B. M., and Kalberer, M.: Online molecular characterisation of organic aerosols in an atmospheric chamber using extractive electrospray ionisation mass spectrometry, Atmos. Chem. Phys., 17, 14485-14500, 10.5194/acp-17-14485-2017, 2017.

Gross, D. S., Galli, M. E., Kalberer, M., Prevot, A. S., Dommen, J., Alfarra, M. R., Duplissy, J., Gaeggeler, K., Gascho, A., Metzger, A., and Baltensperger, U.: Real-time measurement of oligomeric species in secondary organic aerosol with the aerosol time-of-flight mass spectrometer, Anal Chem, 78, 2130-2137, 10.1021/ac060138l, 2006.

Pejov, L.: Metal cations: a density functional, coupled cluser, and quadratic configuration interaction study, International Journal of Quantum Chemistry, 86, 356-367, 10.1002/qua.10022, 2002.

Qi, L., Chen, M.-D., Stefenelli, G., Pospisilova, V., Tong, Y.-D., Bertrand, A., Hueglin, C., Rigler, M., Ge, X.-L., Baltensperger, U., Prévôt, A. S. H., and Slowik, J. G.: Real-time source quantification of wintertime secondary organic aerosol in Zurich using an extractive electrospray ionization time-of-flight mass spectrometer (EESI-TOF), Atmos. Chem. Phys., 19, 8037-8062, 10.5194/acp-19-8037-2019, 2019.

Rodgers, M. T., and Armentrout, P. B.: Absolute binding energies of sodium ions to short chain alcohols, $C_nH_{2n+2}O$, n = 1-4, determined by threshold collision-induced dissociation experiments and ab initio theory, J. Phys. Chem. A, 103, 4955-4963, 10.1021/jp990656i, 1999.

Stefenelli, G., Lopez-Hilfiker, F. D., Pospisilova, V., Vogel, A., Hüglin, C., Baltensperger, U., Prévôt, A. S. H., and Slowik, J. G.: Source apportionment of ambient organic aerosol by online extractive electrospray ionization time-of-flight mass spectrometry (EESI-TOF), Atmos. Chem. Phys. Discuss., 10.5194/acp-2019-361, 2019.